# Brown Rice Vinegar as an Olfactory Field Attractant for *Drosophila suzukii* (Matsumura) and *Zaprionus indianus* Gupta (Diptera: Drosophilidae) in Cherimoya in Maui, Hawaii, with Implications for Attractant Specificity between Species and Estimation of Relative Abundance

**DOI:** 10.3390/insects10030080

**Published:** 2019-03-20

**Authors:** Brittany N. Willbrand, Douglas G. Pfeiffer

**Affiliations:** Department of Entomology, Virginia Tech, 205C Price Hall, Blacksburg, VA 24061, USA; dgpfeiff@vt.edu

**Keywords:** drosophilids, invasive species, attractant effectiveness, field trapping, brown rice vinegar, apple cider vinegar, red wine

## Abstract

*Drosophila suzukii* (Matsumura) is an agricultural pest that has been observed co-infesting soft-skinned fruits with *Zaprionus indianus* Gupta. The characterization of olfactory preferences by species is a necessary step towards the development of species-specific attractants. Five olfactory attractants were used to survey the populations of two invasive drosophilids in cherimoya in Maui, Hawaii. The attractants used were apple cider vinegar (ACV), brown rice vinegar (BRV), red wine (RW), apple cider vinegar and red wine (ACV+RW; 60/40), and brown rice vinegar and red wine (BRV+RW; 60/40). For *D. suzukii*, BRV+RW resulted in more captures than BRV, ACV, and RW, while ACV+RW resulted in more captures than ACV. No differences were observed between BRV+RW and ACV+RW. BRV had greater specificity in attracting *D. suzukii* compared to ACV, ACV+RW, and RW. For *Z. indianus*, no significant differences were observed in either the mean captures or specificity for any attractant used. Collectively, these findings demonstrate that (1) BRV and BRV+RW are effective field attractants and (2) *D. suzukii* has unique olfactory preferences compared to non-target drosophilids, while (3) *Z. indianus’* preferences do not appear to vary from non-target drosophilids, and (4) the accuracy of relative abundance is impacted by the specificity of the attractants.

## 1. Introduction

International trade and travel has supported the dispersal of many insect species outside of their native range including *Drosophila suzukii* (Matsumura) and *Zaprionus indianus* Gupta. Native to Southeast Asia, *D. suzukii* is now commonly found throughout North America [1,2], South America [3], and Europe [4,5,6]. The first adult specimens in the United States were captured on Oahu and Kauai in 1983 [7], and records have since been published across the U.S. mainland [8]. Similarly, *Z. indianus*, native to the Afrotropics, is now reported throughout North America, South America, Europe, and Asia [9,10]. Like many dipterans, *D. suzukii* and *Z. indianus* play a vital ecological role in nutrient cycling through catalyzing decomposition processes. However, when introduced to a non-native range, temperate climates and the abundance of host plants paired with the lack of natural predators can contribute to unregulated population expansion [11]. Furthermore, these vinegar fly species possess high adaptability [12,13] and fecundity [14,15] that facilitate successful colonization. Not surprisingly, *D. suzukii* and *Z. indianus* are considered invasive agricultural pests for many soft-skinned fruits due to increased pest management costs, reduced yields and rejected product [16,17]. 

The pest potential of *D. suzukii* and *Z. indianus* can be attributed to their polyphagous nature [18], transmission of microorganisms to host plant tissues [19], and niche adaptations that enable larval development in ripening fruit [20]. As a polyphagous drosophilid, *D. suzukii* uses a wide range of host plants for feeding and breeding, including nectarines, peaches, plums, pluots, persimmons, figs, strawberry, blackberry, blueberry, raspberry, cherry, and grapes [18,21]. Unlike most vinegar flies that oviposit into overripe and decaying fruits, the evolution of a serrated ovipositor and novel behavioral adaptations enable *D. suzukii* to target undamaged ripening fruit [22]. Similarly, *Z. indianus* has been observed targeting ripening figs [23] and undamaged ripe strawberry [24] as oviposition substrates. Oviposition injury can damage fruit by inoculating the surface with microorganisms that promote decay [25,26] and through larval consumption of fruit pulp post-emergence [24,27]. Eventually, when farmers confirm *D. suzukii* or *Z. indianus* in the field, additional labor costs are warranted for surveying, pest management, and post-harvest handling [17,28]. 

Current field trapping strategies include the use of visual and olfactory attractants, neither of which exclusively attract *D. suzukii* [29] or *Z. indianus* [30]. Sticky cards can serve as visual attractants, particularly fluorescent red or yellow colors [31]. Vinegars and wines serve as olfactory attractants in the field [32,33] that take advantage of drosophilids’ innate attraction to fermentation by-products [34,35,36]. Since there are no species-specific attractants for drosophilids available, farmers must learn to distinguish *D. suzukii* and *Z. indianus* from other insects that are captured in field survey traps. Unfortunately, this practice is not only laborious, but also results in the capture of non-target species, including beneficial parasitoids. Thus, there is crucial need for the development of species-specific attractants. 

The concept of targeted attractants for drosophilid species is plausible as fermentation products are innately attractive to drosophilids [34,35,36], and drosophilids exhibit differences in olfactory preferences. These species-specific preferences to different yeasts [37] may be related to evolutionary adaptations for survival against competing drosophilids over finite resources [23,38], which may eventually result in speciation [39,40,41]. For *D. suzukii*, preferential attraction may be related to physiological adaptations that encourage oviposition in ripening fruit [38], while larval nutritional requirements may have changed over time to permit *Z. indianus* to oviposit in ripe figs [23]. This suggests that preferential attraction to different yeast communities may permit co-habitation within a given environment [37]. *Drosophila suzukii* is attracted not only to ripening fruit [42] and yeasts [37], but also to odors emitted from leaf tissue. After olfactory-mediated arrival to the plant, additional environmental cues may prompt subsequent behaviors such as feeding, breeding, or oviposition [42]. Furthermore, the odors emitted from leaf tissue and fruits will vary between plant species, enabling insect species to co-evolve with particular plants. Research has found that the ratio of protein to carbohydrates (P:C) in a substance affects adult fly behavior; for example, a low P:C ratio was associated with higher oviposition behaviors in *D. suzukii*, which is thought to be an evolutionary trait that enhances survivability in adult females [43]. These preferences, once fully characterized, can be used for targeted field surveillance, attract-and-sterilize, or attract-and-kill strategies. 

The headspace from fermented solutions that are attractive to drosophilids, such as vinegars, wines, and yeast [30,44], can be analyzed to identify the volatile composition [45]. Individual volatiles that are suspected to be attractive can then be tested in laboratory choice assays or in the field to develop targeted bait. Some progress has already been made towards the development of synthetic drosophilid bait [46] for *D. suzukii* [47] and *Z. indianus* [30], but the identification of novel fermenting solutions that are attractive to particular drosophilids and the characterization of these species–attractant interactions can further advance the development of these species-specific attractants. 

Since the volatile composition varies between vinegar products [48], examining the efficacy of novel vinegar products [30] and identifying species-specific preferences may lead to improvements in synthetic attractant development. Previous research has demonstrated that apple cider vinegar [49,50] and apple cider vinegar and red wine [51] are effective field attractants for *D. suzukii* and *Z. indianus*. Rice vinegar, with and without red wine, has also been characterized by several researchers [30,32,46]. Recently, Akasaka et al. [52] demonstrated that brown rice vinegar was more attractive to *D. suzukii* than apple cider vinegar in laboratory choice assays, substantiating the need for the characterization of attractant efficacy in field settings.

The primary objective of this experiment was to examine attractant efficacy and specificity by attractant type (apple cider vinegar, brown rice vinegar, red wine, apple cider vinegar blended with red wine, brown rice vinegar blended with red wine) and species (*Drosophila Suzukii* (Matsumura), *Zaprionus Indianus* Gupta, non-target drosophilids) in cherimoya (*Annona cherimola* Miller). The secondary objective was to examine whether attractant efficacy varied over time in the field capture of *D. suzukii*, *Z. indianus*, and non-target drosophilids. The tertiary objective was to characterize the relative abundance of two invasive drosophilids, *D. suzukii* and *Z. indianus*. This report provides the first published characterization of the relative abundance of *Z. indianus* in Maui, Hawaii, since the reported introduction in 2017 [53]. Furthermore, at the time of writing, this report provides the first published quantitative comparison of brown rice vinegar as an attractant solution in a field setting in the United States. 

## 2. Materials and Methods 

### 2.1. Trap Assembly 

The traps used for the following three experiments were a novel modification of the traditional deli cup design (Figure 1). Assembled using a Mason jar (236 mL) and a red Solo cup (473 mL), the jar served to house the attractant-drowning solution, while the Solo cup provided the headspace that enabled passive diffusion of the attractant and fly entry into the trap. To fabricate the trap, the interior of the Solo cup was scoured with sandpaper, rinsed with 70% ethanol, and then coated with epoxy. The silver band from the Mason jar lid was secured inside the Solo cup, as deep as the diameter of the silver band permitted, and was then left to cure for 24 h. A soldering tool (Cold-Heat^®^) with a 5-mm diameter tip was used to gently puncture eight entry holes into the Solo cup—two on each of the four sides. Two of the holes were used in trap installation with nylon rope. The removable trap base permitted simple trap maintenance as the attractant solutions were aliquoted into clean jars weekly which then replaced the spent jars on-site. After the lids were affixed to the spent jars, they were transported off-site for specimen preservation in alcohol and quantification.

### 2.2. Attractant Solutions 

Five attractants were used in this experiment: apple cider vinegar (ACV, supplied by Marukan Vinegar Co. Ltd., Kobe, Japan), brown rice vinegar (BRV, supplied by Marukan Vinegar Co. Ltd.), red wine (RW, Oak Leaf Vineyards, Merlot), ACV+RW (blended; 60/40%), and BRV+RW (blended; 60/40%). ACV, RW, and ACV+RW were chosen due to the historical use of vinegars and wine as olfactory baits in field traps [54]. BRV was selected due to recent evidence suggesting BRV is more attractive to *D. suzukii* than ACV in laboratory choice-assays [52]. BRV+RW was prepared to assess the dilution of BRV and synergy with red wine volatiles that had been previously observed with ACV+RW blends [30,32]. Attractants were aliquoted (40 mL per trap) into Mason jars on the morning of trap installation. A drop of unscented dishwashing liquid (Dawn^®^ Ultra Free and Gentle, Procter & Gamble, Cincinnati, OH, USA) was added to each jar using a 3-mL transfer pipette (Karter Scientific, Lake Charles, LA, USA). The addition of a surfactant was necessary to break the surface tension, transforming the attractant into a drowning solution. After the surfactants were added, the Mason jar lids were affixed and ready for transport. 

### 2.3. Trap Installation and Maintenance 

The traps were installed at fruiting height and the attractant solution was replaced weekly. The trap height corresponded with low hanging fruit, as research in foraging ecology has shown higher *D. suzukii* capture rates in traps positioned on the lower canopy [55]. After trap installation, the trap bases were replaced weekly to collect captured specimens and to refresh the attractant solution. Once the specimens were collected, they were separated from the attractant solution using a 1-mm mesh filter and were stored in 70% isopropyl alcohol prior to identification.

### 2.4. Species Identification 

The specimens were classified as *D. suzukii* (male or female), *Z. indianus* (male or female), non-target drosophilid, or other non-target captures. Standard keys were used in the examination of external morphology for *D. suzukii* [56] and *Z. indianus* [57,58]. Taxonomic confirmation for *D. suzukii* was confirmed by Dr. Douglas Pfeiffer while *Z. indianus* confirmation was provided by Dr. Luc Leblanc, University of Idaho, and Dr. Amir Yassin, Muséum d’Histoire Naturelle de Paris. The specimens were classified as non-target drosophilids if the individuals possessed all of the following attributes: (1) a body length of 3–4 mm, (2) a golden-yellow-brownish body color, (3) red eyes, (4) antennae with branched arista, and (5) wings that have three breaks in the costa vein [59]. The specimens that did not meet the criteria for *D. suzukii*, *Z. indianus*, or non-target drosophilid, were classified as non-target other captures. Note that this included the captures of any true fruit fly from Tephritidae, such as *Ceratitis capitata* (Wiedemann), the Mediterranean fruit fly, or *Bactrocera cucurbitae* (Coquillett), the melon fly.

### 2.5. Experimental Design 

The survey location that was selected was a cherimoya grove at the University of Hawaii, at Mānoa: Kula Research Station (20.7° N; −156.3° W, 980 m) (Figure 2). There were 24 traps installed, which included four replicates of the six attractant treatments (five attractants and one control). Distilled water was used to control for a confounding variable—drosophilid attraction to visual stimuli [31], i.e., the red Solo cup. Once the traps were delineated and installed, attractant treatment was randomized. Each week, the captured specimens were collected, trap bases were replaced, and attractant treatment was re-randomized. Thus, each week corresponded to one run or replicate of the experiment. For each run, the four repeats of each attractant level were treated as individual data points and were not pooled or averaged for analysis, as the experimental unit was defined as an individual trap. 

The cherimoya grove at the University of Hawaii, at Mānoa: Kula Research Station, contained fruit that was ripening or ripe. The grove floor was shady with small amounts of plant debris in the early stages of decomposition and weeds that were routinely treated with herbicides. The perimeter of the grove contained approximately 2.5 m of well-managed turfgrass. On 13 November 2017, 24 traps were installed and were maintained weekly until 11 December 2017 (N = 16). Four rows of trees were selected, and six traps were installed within each row. The traps were placed in a single tree with at least 5 m between each trap. 

### 2.6. Statistical Analysis 

Statistical analysis was conducted with SAS^®^ (SAS Institute Inc., Cary, NC, USA), with the statistical model as a generalized linear mixed model (GLMM) (PROC GLIMMIX procedure). A GLMM was selected as the data analyzed were non-normal and not transformed, and the model was subjected to both fixed and random effects. The significance threshold for mean separation was defined as *p* < 0.05 for post-hoc analysis. Tukey’s Honestly Significant Difference (HSD) test was used for mean separation (within the LSMEANS statement). Several resources were utilized in the modeling of count data with a Poisson distribution [60,61,62]. In order to determine whether there were significant differences in the mean captures between the sexes, the total number of *D. suzukii* males was divided by the total number of male and female *D. suzukii* captures.

The sex specificity value results in a single response variable with a value between 0 and 1 for each datapoint (row of the spreadsheet). A value of 1.0 implies that only males were captured, while a value of 0.5 implies that there was equal male to female captures, and a value of 0.0 indicates that only females were captured. 

To examine whether there were sex differences in mean capture between attractants, a generalized linear mixed model (GLMM) was used (PROC GLIMMIX) with the attractant as a fixed effect and with the week and replicate as random effects. The method was set as Laplace, and a beta distribution was specified as the sex specificity calculation was expressed as a decimal fraction. If there were no significant differences in sex specificity between either male and female *D. suzukii* or between male and female *Z. indianus*, further analysis was performed with the sex counts pooled. 

To examine differences in mean capture between attractants, a GLMM was used (PROC GLIMMIX), with the attractant as a fixed effect and with the week and replicate as random effects. The method was set as Laplace, and a Poisson distribution was specified for the raw count measurements for each specimen category (*D. suzukii* female, *D. suzukii* male, *Z. indianus* female, *Z. indianus* male, non-target drosophilid, and other non-targets).

To examine the specificity between *D. suzukii* vs. drosophilid captures and *Z. indianus* vs. drosophilid captures, the raw data (capture counts) were manipulated to result in a single response variable (specificity ratios) on a per trap basis. For each trap, the attractant specificity for *D. suzukii* was calculated by dividing *D. suzukii* captures by drosophilid captures. Similarly, to calculate the attractant specificity for *Z. indianus*, *Z. indianus* captures were divided by drosophilid captures, resulting in a specificity value for each trap. The species specificity calculation results in a value between 0 and 1 for each datapoint (row of the spreadsheet) with the interpretation of values similar to as previously described for sex specificity. Using this calculation, mean specificity by attractant type could be evaluated. 

To examine whether there were differences in the specificity of captures between attractants, a generalized linear mixed model (GLMM) was used (PROC GLIMMIX) with attractant as a fixed effect and with week and replicate as random effects. The method was set as Laplace, and a beta distribution was specified as the specificity results were expressed as decimal fractions. To support the secondary objective, a GLMM was used (PROC GLIMMIX) to examine whether the mean captures or specificity varied by week (fixed effects are attractant, week, and attractant*week; random effect is replicate). Similar to the previous analysis, the method was set as Laplace, and the distribution was set as Poisson and beta for count data and decimal fractions, respectively. To support the tertiary objective, relative abundance ratios were generated by dividing the total quantity of target specimens captured (*D. suzukii* or *Z. indianus*) by the total quantity of vinegar flies captured (*D. suzukii*, *Z. indianus*, and non-target drosophilid). Though similar to the calculation of specificity, the estimation of relative abundance produces a single value which includes all attractant solutions, while specificity is calculated on the level of each subject resulting in 80 values for statistical analysis. 

## 3. Results

### 3.1. Drosophila suzukii: Mean Captures and Attractant Specificity

Since there were no significant differences between male and female *D. suzukii* captures by attractant (f = 0.73, df = 4, 68, *p* = 0.5761), the captures were pooled for both sexes for analysis. There were significant differences in *D. suzukii* captures by attractant type (f = 5.86, df = 4, 75, *p* < 0.001) (Figure 3). Mean separation using Tukey’s HSD test indicated that BRV+RW (29.0 ± 4.4) resulted in more captures than BRV (16.5 ± 3.0), RW (12.8 ± 1.6), and ACV (9.3 ± 2.3). There were no significant differences between BRV+RW (29.0 ± 4.4) and ACV+RW (17.6 ± 3.0). There were no significant differences between ACV+RW (17.6 ± 3.0), BRV (16.5 ± 3.0), and RW (12.8 ± 1.6). The use of ACV+RW (17.6 ± 3.0) or BRV (16.5 ± 3.0) resulted in higher captures than ACV alone (9.3 ± 2.3). 

There were significant differences observed in the mean captures of *D. suzukii* by week surveyed (f = 25.93, df = 3, 60, *p* < 0.0001) (Figure 4). The greatest mean captures of *D. suzukii* occurred in weeks 2 (24.5 ± 2.8) and 3 (22.0 ± 3.4), followed by week 1 (14.3 ± 2.7), with the lowest captures observed in week 4 (7.5 ± 1.1).

Since the interaction between week and attractant was significant (f = 3.43, df = 12, 60, *p* < 0.001), attractant preferences were examined separately for each week. Significant differences were observed in the mean captures by attractant in week 1 (f = 6.13, df = 4, 15, *p* < 0.01), week 2 (f = 6.73, df = 4, 15, *p* < 0.01), week 3 (f = 4.64, df = 4, 15, *p* < 0.05), and week 4 (f = 5.76, df = 4, 15, *p* < 0.01) (Figure 5). For week 1, the use of BRV+RW (33.5 ± 4.3) resulted in more captures than BRV (11.5 ± 5.7), RW (10.0 ± 3.0), ACV+RW (9.5 ± 2.9), and ACV (6.75 ± 1.8). For week 2, the use of BRV+RW (32.0 ± 4.5), ACV+RW (31.8 ± 7.3), BRV (30.0 ± 5.6) or RW (19.3 ± 2.5), resulted in more captures than ACV (9.5 ± 3.5). Week 3 was similar to week 1—the use of BRV+RW (43.8 ± 10.1) resulted in more captures than ACV (19.3 ± 5.7), ACV+RW (16.8 ± 4.5), BRV (16.5 ± 3.2), and RW (13.5 ± 2.5). In contrast, for week 4, ACV+RW (12.5 ± 1.3) resulted in more captures than BRV+RW (6.75 ± 1.6) and ACV (1.75 ± 0.5). The use of RW (9.25 ± 1.9), BRV (8.0 ± 2.7), or BRV+RW (6.75 ± 1.6) resulted in more captures than ACV (1.75 ± 0.5). There were no significant differences between ACV+RW (12.5 ± 1.3), RW (9.25 ± 1.9), and BRV (8.0 ± 2.7). There were no significant differences between RW (9.25 ± 1.9), BRV (8.0 ± 2.7), and BRV+RW (6.75 ± 1.6). 

There were significant differences observed in the specificity of *D. suzukii* captures per trap by attractant (f = 4.39, df = 4, 74, *p* > 0.01) (Figure 6). The use of BRV alone (0.47 ± 0.05) resulted in higher specificity than ACV (0.38 ± 0.06), ACV+RW (0.32 ± 0.04), and RW (0.26 ± 0.03). There was no significant difference in specificity for *D. suzukii* between BRV (0.47 ± 0.05) and BRV+RW (0.41 ± 0.04). There were no significant differences in specificity for *D. suzukii* between BRV+RW (0.41 ± 0.04) and ACV+RW (0.32 ± 0.04).

### 3.2. Zaprionus indianus: Mean Captures and Attractant Specificity 

Since there were no significant differences between male and female *Zaprionus indianus* captures by attractant (f = 1.43, df = 4, 14, *p* = 0.2762), the captures were pooled for both sexes for analysis. No significant differences were observed for *Zaprionus indianus* captures by attractant type (f = 1.41, df = 4, 75, *p* = 0.2376) (Figure 7). 

There were no significant differences observed in the mean captures of *Z. indianus* by week (f = 2.74, df = 3, 60, *p* = 0.0511); the interaction between attractant and week was also non-significant (f = 1.81, df = 12, 60, *p* = 0.0671) (Figure 8). 

There were no significant differences observed in the specificity of *Z. indianus* captures per trap by attractant (f = 1.82, df = 4, 37, *p* = 0.1450) (Figure 9).

### 3.3. Non-Target Drosophilid: Mean Captures and Attractant Specificity 

There were significant differences in the mean captures of non-target drosophilids by attractant (f = 8.63, df = 4, 75, *p* < 0.0001) (Figure 10). The use of RW (43.8 ± 6.7), BRV+RW (41.7 ± 7.5), and ACV+RW (39.1 ± 5.8) resulted in more captures than BRV (18.4 ± 3.6) and ACV (13.1 ± 2.0) alone.

There were significant differences observed in the mean captures by week surveyed (f = 37.25, df = 3, 60, *p* < 0.0001) (Figure 11). The greatest mean captures of non-target drosophilids occurred in week 3 (52.8 ± 6.8), followed by week 2 (36.4 ± 4.4), week 1 (23.0 ± 4.6), and the lowest were observed in week 4 (12.7 ± 1.7). The interaction between week and attractant was not significant (f = 1.44, df = 12, 60, *p* = 0.1748). 

### 3.4. Estimated Relative Abundance of D. suzukii, Z. indianus, and Non-Target Drosophilids in Cherimoya 

To characterize the relative abundance of exotic drosophilids, total captures by species and attractant type were reported in a contingency table (Table 1), then the relative abundance of *D. suzukii*, *Z. indianus*, and non-target drosophilids was calculated (Figure 12). All experimental data, including raw counts and calculated specificity ratios, can be found in the Appendix A.

## 4. Discussion

The results of this experiment for the field capture of drosophilids in cherimoya in Maui suggest that (1) BRV and BRV+RW are effective attractants for *D. suzukii* and *Z. indianus,* (2) *D. suzukii* has unique olfactory preferences compared to all non-target drosophilids, while (3) there are no differences in olfactory preferences between *Z. indianus* and non-target drosophilids, (4) attractant preferences vary over time for *D. suzukii* but not for *Z. indianus*, and (5) the accuracy of relative abundance is impacted by the specificity of the attractants used. At the time of writing, this is the first reported use of BRV and BRV+RW as olfactory attractants in a quantitative field trapping study in the United States. Furthermore, this report provides the first characterization of the relative abundance of *Z. indianus* since its reported introduction in Maui [53]. Since there were relatively few captures in the control treatment (distilled water), which controlled for drosophilid attraction towards visual stimuli (to the red solo cup) [31], it can be concluded that the captures in this experiment were due to olfactory attraction or the combination of olfactory and visual stimuli—not to visual stimuli alone. The limitations of this experiment include limited duration (4 weeks) and relatively low repeats (N = 16), suggesting that further research is needed for greater confidence in these trends. Nonetheless, this experiment provides insight into attractant efficacy by species and emphasizes the need to assess the synergy between attractant solutions and other compounds within the host plant environment, such as the outgassing of volatiles emitted from plants and fruit, which has been a topic of recent investigation [63,64]. Furthermore, since the experiment was limited to one host plant on one island, additional experiments in other host plants and localities are needed to have confidence in extrapolating these trends across multiple environments.

Five attractants were used (ACV, BRV, RW, ACV+RW, and BRV+RW) to evaluate attractant effectiveness for *D. suzukii* and *Z. indianus* in cherimoya. There were significant differences in the mean captures of *D. suzukii* by attractant type, but no significant differences were observed for *Z. indianus*. The use of BRV+RW resulted in greater *D. suzukii* captures than BRV, RW, and ACV. There were no significant differences between BRV+RW and ACV+RW, suggesting that BRV+RW may be equivalent to ACV+RW as an olfactory attractant for *D. suzukii*. Similar to the results of previous experiments, the use of ACV+RW resulted in greater *D. suzukii* captures than ACV alone [32,46]. Researchers conducting field experiments in berry crops found that wine-vinegar blends were more attractive to *D. suzukii* than wine and vinegar alone [46] or aqueous red wine [32]. Cha et al. [46], reported that blended RW and rice vinegar (60/40%) was more attractive to *D. suzukii* than rice vinegar or red wine alone, while Landolt et al. [32] found that RW blended with rice vinegar (60/40%) was more attractive to *D. suzukii* than RW with ACV (60/40%) or RW with white wine vinegar (60/40%) or aqueous RW (60%). In contrast to the previous experiments described, there was no significant difference between ACV+RW and RW, which could be attributed to other factors present between the different host plant environments examined—cherimoya in this experiment vs. blackberry [32] and berry crops [46]. For *Z. indianus*, no significant differences were observed in the mean captures for any attractant used, which suggests that, in practice, any of the attractants assessed could be used to identify whether *Z. indianus* adults are present in a field. For non-target drosophilids, greater captures were observed with RW, ACV+RW, and BRV+RW than with ACV or BRV alone. This suggests that for non-target drosophilids and *Z. indianus*, wine volatiles alone performed similarly to the synergy of wine and vinegar volatiles, while the latter was a stronger preference for *D. suzukii*.

There were significant differences in the mean captures by week for *D. suzukii* and non-target drosophilids, but not for *Z. indianus*. After analysis of *D. suzukii* attractant preferences by week, there appeared to be slight differences in the distribution of the captures. For non-target drosophilids, even though the captures varied by week, attractant preferences were similar throughout the duration of the experiment. Inclusive of all attractant types examined, 65% of total *D. suzukii* captures were male and 35% were female. Since there were no significant differences in the proportion of male to female captures by attractant type, this suggests that the trend was consistent regardless of which attractant was used with 60–67% male captures to 33–40% female captures. Though it is not possible to know the actual sex ratio for the population and since the visitor rate was not assessed (e.g., pesticide treatment of traps coupled with a bucket), it is not possible to determine whether there were more males present in the population or simply more male captures.

In order to develop species-specific attractants, the characterization of attractant specificity by species is needed. BRV had greater specificity in attracting *D. suzukii* compared to ACV, ACV+RW, or RW. On average, ~50% of the drosophilid captures in the BRV traps were *D. suzukii*, which was reduced to ~40% with BRV+RW and ACV, ~30% with ACV+RW, and ~25% with RW. The results of this field experiment support previous research where, in a laboratory trapping assay, BRV was found to be significantly more attractive to *D. suzukii* than ACV [52]. Though there were no significant differences in specificity observed between BRV and BRV+RW, there were also no differences between BRV+RW and ACV+RW, suggesting that the volatile compound present in BRV, once diluted with red wine, becomes less specific to attracting *D. suzukii*. Since the results suggest that BRV has greater *D. suzukii* specificity, it follows that using BRV as a field attractant for surveying could reduce non-target impacts and the time necessary to identify specimens. However, further research in additional host plants and localities is needed to better understand this trend and identify any potential ecological and economic impacts. The specificity reported in this experiment was calculated from observed captures and did not take into account the true population, as this was an open-field experiment. Future research could examine attractant specificity between *D. suzukii* and *Z. indianus* in a controlled environment, which may better isolate the variables in question and provide valuable data towards the development of a species-specific attractant. The preferential attraction observed in this experiment is likely associated with evolutionary adaptations that enable co-habitation of *D. suzukii* and *Z. indianus* with other drosophilids over finite resources in a given environment [23,41,42]. These unique olfactory preferences have implications for both farmers and researchers. In order to mitigate revenue loss, farmers must detect the presence of pests as early in the season as possible to start implementing pest reduction strategies. These findings suggest that for field surveying drosophilid pests, multiple attractants should be used to cover a range of attractive compounds.

The results of this experiment beg the question—if specificity varies depending on attractant, how can an accurate representation of relative abundance be depicted from the field captures of wild flies? In this experiment, the relative abundance of *D. suzukii* and *Z. indianus* ranged from 22.3–46.4% and 1.1–5.8%, respectively, depending on the attractant used. The low relative abundance of *Z. indianus* observed could be the result of low attractant efficacy in the host plant environment, or that cherimoya was not a preferred host—both of which may have underestimated the actual population. Alternatively, the low relative abundance could correspond with the recent introduction of this species to the region [53] coupled with interspecies competition for resources that limited population expansion. In central Brazil, Tidon et al. [65] reported that *Z. indianus* had an initially low relative abundance followed by a population explosion the following year when the populations were re-surveyed. Since the actual relative abundance cannot be known in wild populations, and there is attractant bias, the interpretation of field data on relative abundance is challenging and may overestimate or underestimate the actual population. This experiment emphasizes the challenges and limitations associated with the interpretation of relative abundance calculations from field-generated data. 

Previous research has demonstrated the potential for co-infestation of *D. suzukii* and *Z. indianus* in a variety of crops including strawberry [24], guava [2] and grape [66]. As a secondary pest with opportunistic oviposition tendencies, *Z. indianus* can increase damage to soft-skinned fruits with oviposition injury from *D. suzukii* [66] or mechanical damage from other insects [24]. In Veracruz, Mexico, *D. suzukii* and *Z. indianus* co-infested guava orchards comprising more than 80% of the total drosophilids captured (55.0 and 26.3% relative abundance, respectively) [2], which suggests that there may be deleterious interspecies competition between endemic and invasive drosophilids. There is also evidence of interspecies competition between *D. suzukii* and *Z. indianus*. For example, larval competition between *Z. indianus* and *D. suzukii* increased development time and mortality for *D. suzukii* in Virginia [66]. Even though co-infestation was observed in the cherimoya grove, no attempt was made to examine interspecies competition or to quantify the losses from direct and indirect injury. Furthermore, research is needed regarding *D. suzukii* and *Z. indianus* co-infestation to identify economic thresholds and provide practical pest management strategies for farmers.

Several limitations were identified in this experiment including the short duration of the study. The short-duration of the experiment (4 weeks) limited the characterization of olfactory preferences by fruiting stage over time. Over the four-week experiment, there were few total captures for traps treated with distilled water—three *D. suzukii* specimens were captured (one male, two female), zero *Z. indianus* specimens were captured, five non-target drosophilids were captured, and nine non-target other specimens were captured. Due to the low quantity of the control captures, there is confidence that the dataset analyzed contained values that were the result of olfactory attraction and not confounded by visual stimuli. Additional research is necessary to assess whether an attractant is more or less attractive in different visual environments. Further characterization of attractant specificity by species is needed as fruit ripeness, population dispersal, and interspecies competition alter drosophilid species composition throughout the growing season. Future research could assess attractant efficacy by host plant fruiting stage by collecting fruit samples weekly and identifying the outgassing fruit volatiles. If novel compounds or the synergistic effects of multiple compounds were identified, this would aid in the development of a species-specific attractant. In the meantime, this may also provide farmers with the most effective attractant to use while surveying depending on host plant fruiting stage.

## 5. Conclusions

The results of this study support the notion that *D. suzukii* displays unique olfactory preferences compared to non-target drosophilids, whereas no attractant preferences were observed between *Z. indianus* and non-target drosophilids. The results suggest that BRV, or BRV+RW, is an effective alternative to ACV or ACV+RW for the field survey of drosophilid pests. Furthermore, BRV appears to have high specificity for *D. suzukii* as compared to non-target drosophilids, which warrants further research towards the development of a species-specific attractant. This study provides an initial characterization of the relative abundance of *Z. indianus* (inclusive of all attractants, 2.9%) in a cherimoya grove in Kula, Maui, Hawaii, since the reported introduction in 2017. However, the estimation of relative abundance varied by attractant type (between 1.1–5.8%) and may have been underestimated as none of the attractants used in the experiment had high specificity for *Z. indianus* field captures. Alternatively, the low relative abundance observed could be indicative of a low population level. The actual drosophilid distribution can only be estimated, as the true population is unknown.

## Figures and Tables

**Figure 1 insects-10-00080-f001:**
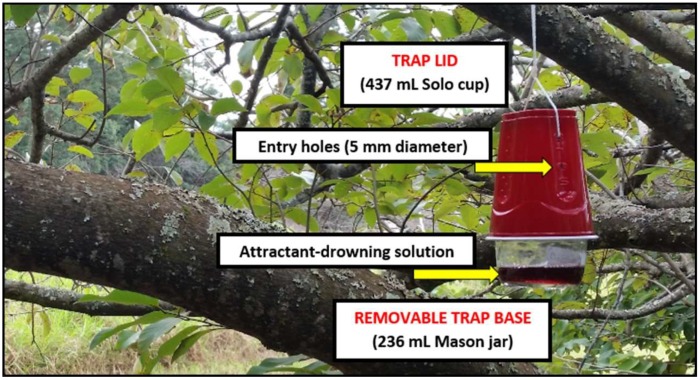
Assembled trap hanging in a cherimoya tree.

**Figure 2 insects-10-00080-f002:**
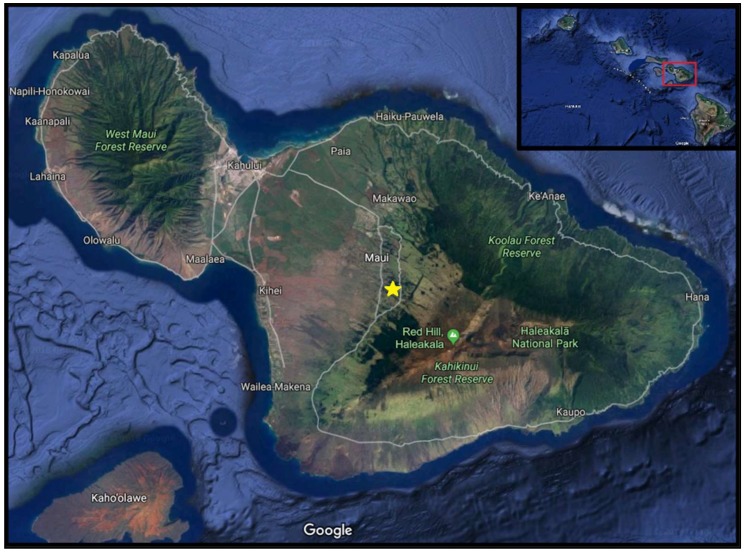
Satellite view of Maui with survey locality designated with yellow star.

**Figure 3 insects-10-00080-f003:**
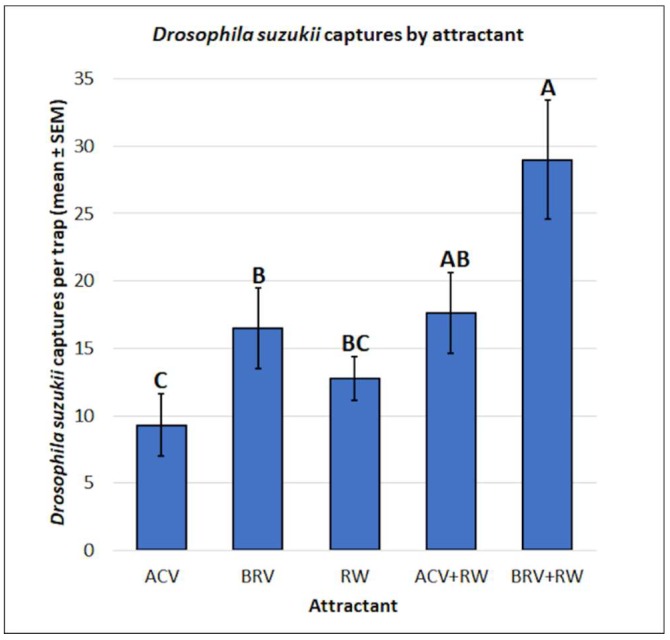
Mean (± SEM) *Drosophila suzukii* captures by attractant type. Different letters represent statistically significant differences (*p* < 0.05, Tukey’s HSD test) between attractant solutions.

**Figure 4 insects-10-00080-f004:**
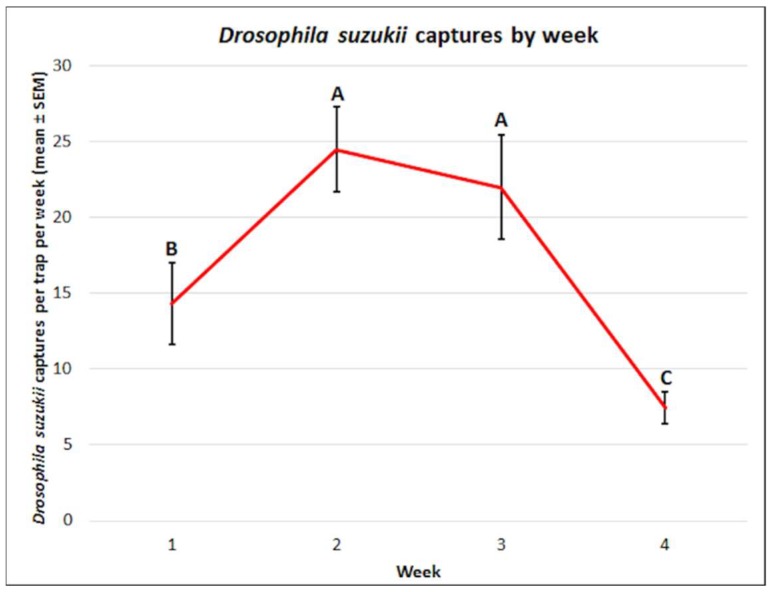
Mean (± SEM) *Drosophila suzukii* captures by week. Different letters represent statistically significant differences (*p* < 0.05, Tukey’s HSD test) between attractant solutions.

**Figure 5 insects-10-00080-f005:**
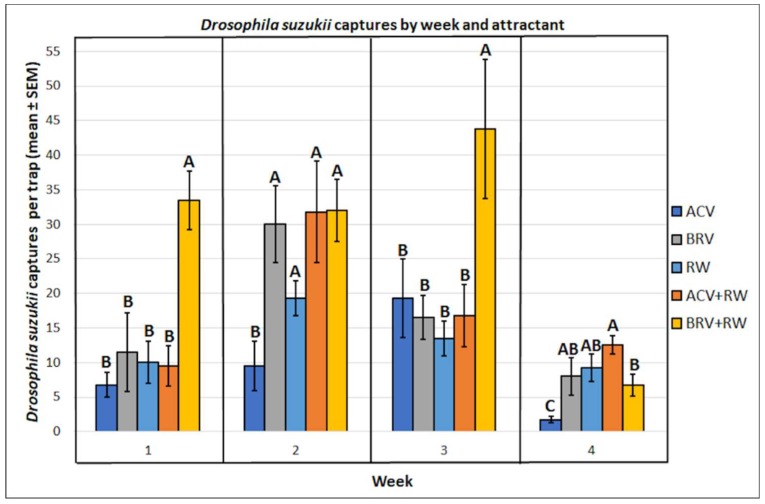
Mean (± SEM) *Drosophila suzukii* captures per trap by week and attractant. Different letters represent statistically significant differences (*p* < 0.05, Tukey’s HSD test) between attractant solutions.

**Figure 6 insects-10-00080-f006:**
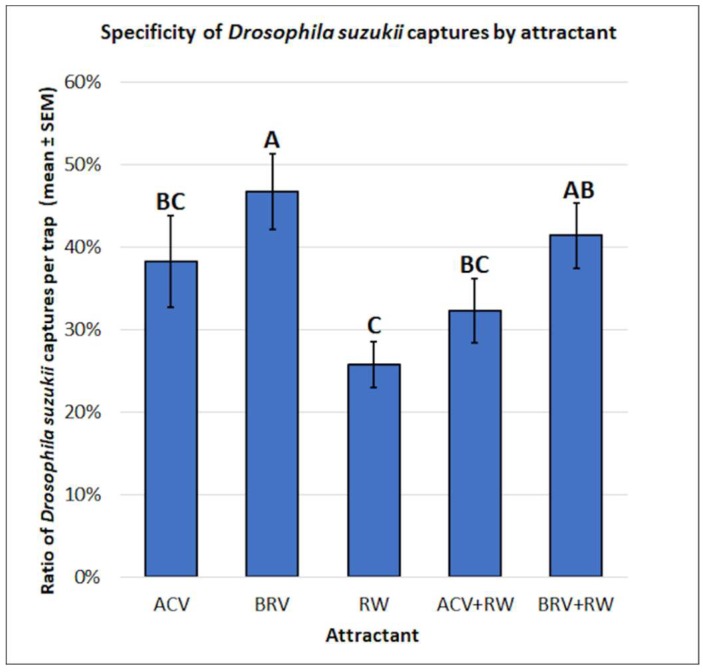
Specificity of *Drosophila suzukii* captures by attractant. Different letters represent statistically significant differences (*p* < 0.05, Tukey’s HSD test) between attractant solutions.

**Figure 7 insects-10-00080-f007:**
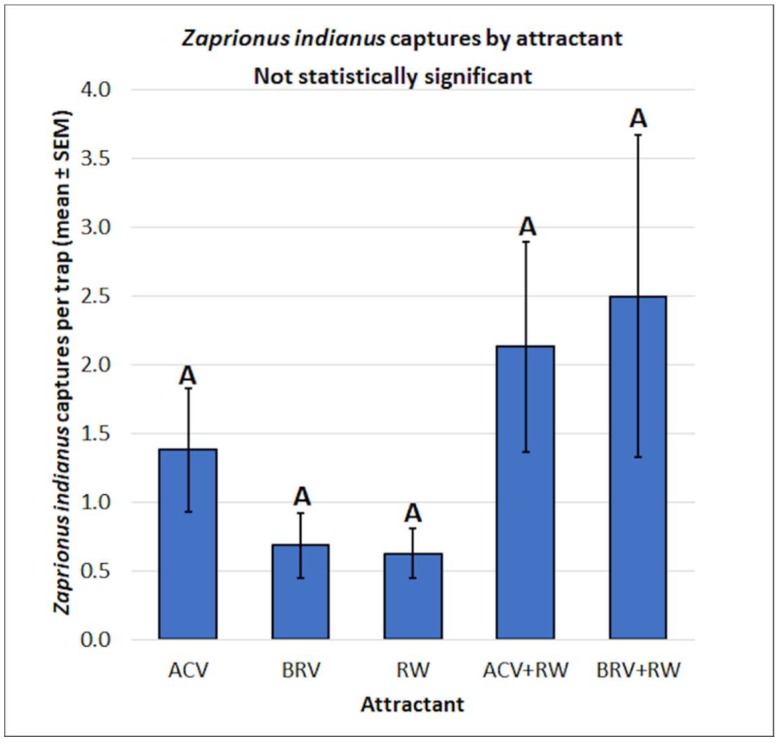
Mean (± SEM) *Zaprionus indianus* captures by attractant type. Different letters represent statistically significant differences (*p* < 0.05, Tukey’s HSD test) between attractant solutions.

**Figure 8 insects-10-00080-f008:**
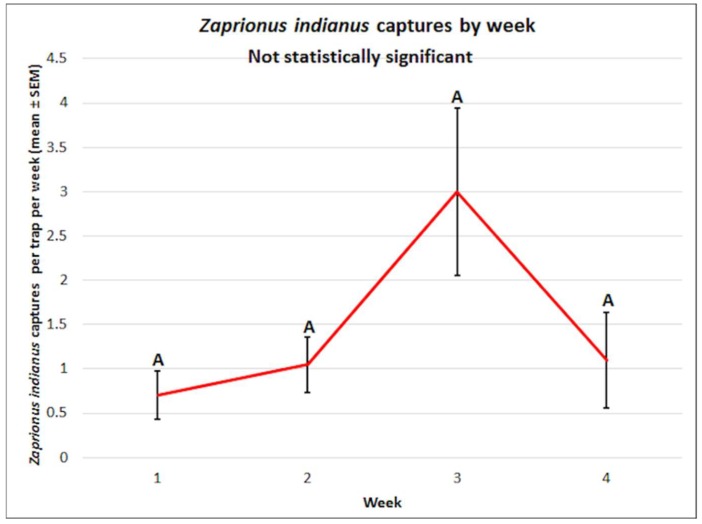
Mean (± SEM) *Zaprionus indianus* captures by week. Different letters represent statistically significant differences (*p* < 0.05, Tukey’s HSD test) between attractant solutions.

**Figure 9 insects-10-00080-f009:**
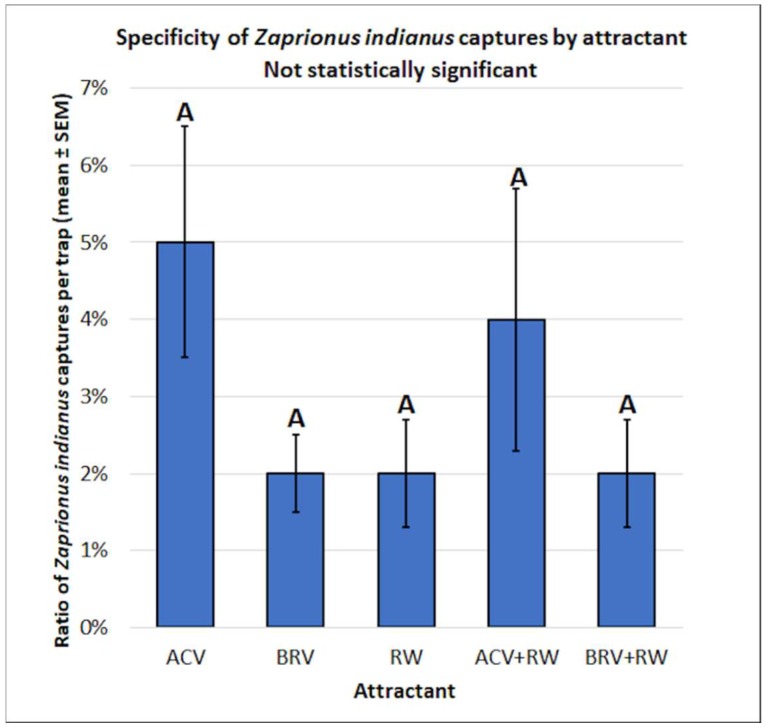
Specificity of *Zaprionus indianus* captures by attractant. Different letters represent statistically significant differences (*p* < 0.05, Tukey’s HSD test) between attractant solutions.

**Figure 10 insects-10-00080-f010:**
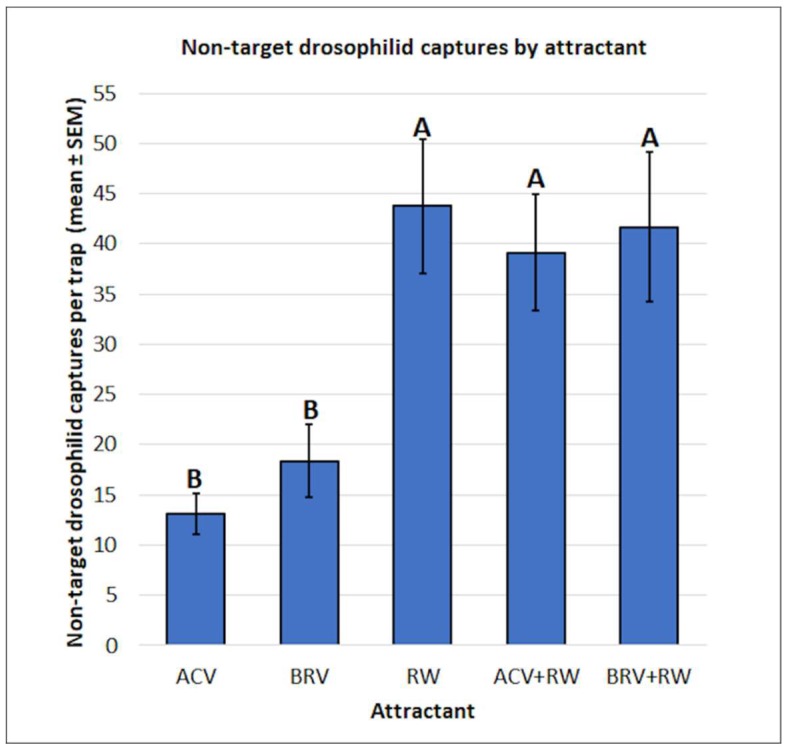
Mean (± SEM) non-target drosophilid captures by attractant. Different letters represent statistically significant differences (*p* < 0.05, Tukey’s HSD test) between attractant solutions.

**Figure 11 insects-10-00080-f011:**
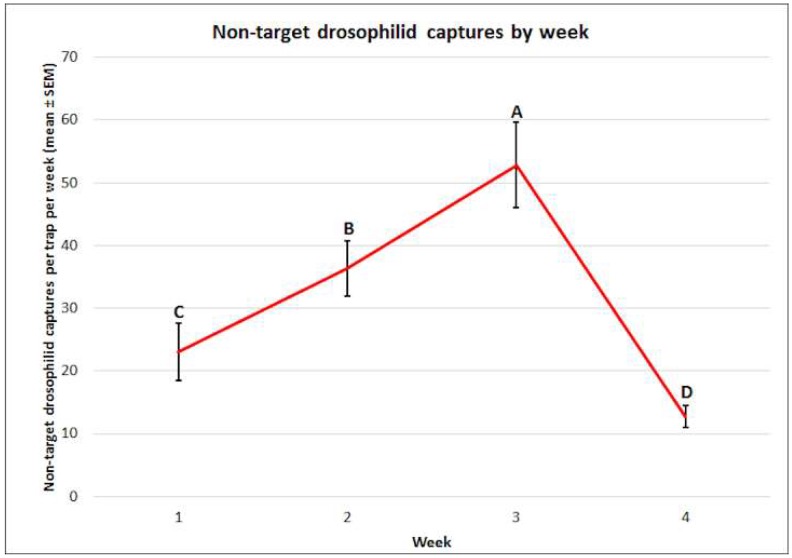
Mean (± SEM) non-target drosophilid captures by week. Different letters represent statistically significant differences (*p* < 0.05, Tukey’s HSD test) between attractant solutions.

**Figure 12 insects-10-00080-f012:**
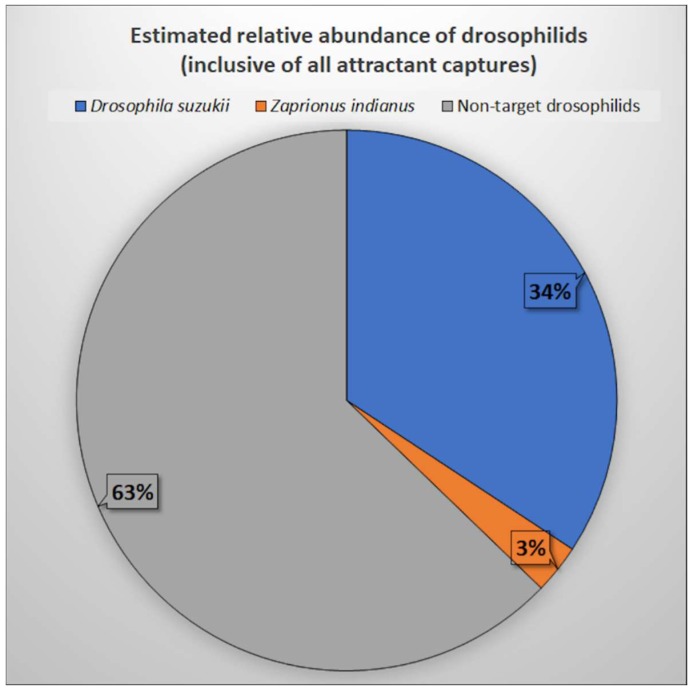
Estimated relative abundance of drosophilids.

**Table 1 insects-10-00080-t001:** Contingency table listing all field captures by attractant type and specimen category.

Attractant Type	*Drosophila suzukii* Male	*Drosophila suzukii* Female	*Zaprionus indianus* Male	*Zaprionus indianus* Female	Non-Target Drosophilids	Non-Target Other	Total Captures
Apple cider vinegar	99	50	14	8	210	11	392
Brown rice vinegar	173	91	8	3	294	21	590
Red wine	123	81	5	5	700	76	990
Apple cider vinegar + red wine	190	92	12	22	626	59	1001
Brown rice vinegar + red wine	300	164	21	19	667	51	1222
Distilled water	1	2	0	0	5	9	17
Total	886	480	60	57	2502	227	4212

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
