# Peer review of "Brown Rice Vinegar as an Olfactory Field Attractant for Drosophila suzukii (Matsumura) and Zaprionus indianus Gupta (Diptera: Drosophilidae) in Cherimoya in Maui, Hawaii, with Implications for Attractant Specificity between Species and Estimation of Relative Abundance"

_insects, 2019, doi:10.3390/insects10030080_

Round 1

Reviewer 1 Report

The objective of this study were to evaluate how the attractiveness of different baits to SWD and Zaprionus indianus differs in different host-plant environments and to show the relative abundance of each of these pest species in three different host plant environments. The purpose for this study was to shed light on olfactory preferences of two crop pests commonly collected in the same environments in order to develop species-specific baits. The question of bait specificity among these two pests is important since they tend to be captured in the same traps but in different numbers and appear to share host plants. The authors evaluated several food-based baits from early literature on SWD but also include brown rice vinegar, which has not been evaluated before.

The introduction and methods sections are extremely detailed and mostly easy to understand. The statistical analysis should be redone using the appropriate methods (details below). I suggest reviewing the questions of the study and making sure the data analysis is appropriate to answer those questions. The authors do not compare among the environments so do not address the question in their title. Also, the number of SWD and Zaprionus are not compared to each other for each bait so the question of bait specificity is not addressed.

Major comments:

L 27 – keywords should not also appear in the title of the paper.

Introduction

The introduction could use some discussion on how and why olfactory responses would differ among different environments.

Experiment sites (crop environments)

Line 92 – why did the authors choose the environments that they did (cherimoya, lemon, strawberry)? Are these two flies pests in each of those crops and therefore are expected to be captured at each location? I am not familiar with cherimoya but I imagine the rind of a lemon would be too thick and tough for either fly to penetrate without the lemon being damaged. Strawberry is an important host plant of SWD and as the authors mention, can be for Zaprionus. However, the study was conducted post-harvest when fruit present was over-ripe. If SWD and Zaprionus prefer ripe strawberries (Bernardi et al. 2017) then would the authors expect for these traps to give an adequate picture of olfactory response post-season?

Replication over time ranged from 2 (lemon and strawberry) to 4 (cherimoya). Attractiveness of baits can vary over the season due to changes in nutritional needs and reproductive status of the flies. Based on the statistical models in this study, captures may have varied by attractant over time. Once the analysis is redone with the appropriate models the authors may want to evaluate whether 2 replications over time in lemon and strawberry are adequate and potentially omit those two environments.

Statistics

I have some major concerns with the statistical model that was used. Based on the text, the authors used a two-way ANOVA with attractant, week(block), and attractant*week(block) as the fixed effects.

1.      First, unless the authors are interested in the effect of the attractant over time (the interaction) then the week(block) should be added as a random variable rather than a fixed variable. If the authors are interested in the effect of attractant over time then it would be considered a longitudinal study whereby each count was taken on the same trap and would need to be analyzed using repeated measures. Having only two time points in lemon and strawberry however, may not provide enough data for repeated measures analysis.

2.      Next, why wasn’t replicate included as a random variable in the model? This would account for any variability that may be a result of the replication over space. This would also address what I believe the authors were testing for in lines 188-192, of positional bias of the traps within each environment.

3.      Finally, there have been a multitude of studies suggesting that count data is better modelled using a generalized linear model (PROC GLIMMIX for generalized linear mixed models in SAS). They are not new but are becoming the standard for modelling this type of data. The distribution function is defined in the model (eg. Poisson or Negative Binomial for count data) so no transformation is required. Additionally, the long final paragraph in the statistics section of the paper would not be necessary. See Bolker et al. 2009 https://doi.org/10.1016/j.tree.2008.10.008; Gbur et al. 2012 https://dl.sciencesocieties.org/publications/books/tocs/acsesspublicati/analysisofgener for starters

Additionally, there is no statistical comparison between male and female captures of SWD or Zaprionus or between SWD and Zaprionus. The authors cannot state that the results of their study suggest that these two pest species have unique olfactory preferences (lines 364-366) if they were not compared. There was also no comparison among the different environments.

Author Response

Reviewer 1 Comments

Summary of manuscript: The objective of this study were to evaluate how the attractiveness of different baits to SWD and Zaprionus indianus differs in different host-plant environments and to show the relative abundance of each of these pest species in three different host plant environments. The purpose for this study was to shed light on olfactory preferences of two crop pests commonly collected in the same environments in order to develop species-specific baits. The question of bait specificity among these two pests is important since they tend to be captured in the same traps but in different numbers and appear to share host plants. The authors evaluated several food-based baits from early literature on SWD but also include brown rice vinegar, which has not been evaluated before.

Summary of comments: The introduction and methods sections are extremely detailed and mostly easy to understand. The statistical analysis should be redone using the appropriate methods (details below). I suggest reviewing the questions of the study and making sure the data analysis is appropriate to answer those questions. The authors do not compare among the environments so do not address the question in their title. Also, the number of SWD and Zaprionus are not compared to each other for each bait so the question of bait specificity is not addressed.

Major comments:

Point 1: L 27 – keywords should not also appear in the title of the paper.

Response 1: Removed Hawaii, Drosophila suzukii, and Zaprionus indianus from the title page of the paper.

Introduction

Point 2: The introduction could use some discussion on how and why olfactory responses would differ among different environments.

Response 2: To clarify how and why olfactory responses may differ among different environments and species, the following text was added (added text in bold) to the introduction:

“The concept of targeted attractants for drosophilid species is plausible as fermentation products are innately attractive to drosophilids [34-36] and drosophilids exhibit differences in olfactory preferences. Genetic and epigenetic traits regulate behavior and, over-time, as drosophilid populations compete over finite resources, these behaviors either result in no viable offspring, have no effect on viable offspring, or enhance the success and/or quantity of offspring. Generally, behaviors that are associated with no viable offspring will eventually become extinct in a population, while behaviors that have no effect or a positive effect on offspring will remain as traits within the population. Over-time, particularly with geographic isolation, these ecological shifts can dramatically shape the behavior, morphology, and genetics of populations. Often, research of these mechanisms is performed retroactively, after speciation, through a variety of methods, such as examination of olfactory preferences [37] behavioral experiments, and molecular analysis [38-40]. These species-specific preferences to different yeasts [41] may be related to evolutionary adaptations for survival against competing drosophilids over finite resources [23,42]. For D. suzukii, preferential attraction may be related to physiological adaptations that encourage oviposition in ripening fruit [42] while larval nutritional requirements may have changed over time to permit Z. indianus to oviposit in ripe figs [23]. This suggests that preferential attraction to different yeast communities may permit co-habitation within a given environment [41]. Not only is D. suzukii preferentially attracted to ripening fruit [37] and yeasts [41] but also the odors emitted from leaf tissue, which is thought to attract the flies to plant where additional environmental cues may result in additional behaviors such as feeding, breeding, or oviposition [37]. Furthermore, the odors emitted from leaf tissue and fruits will vary between plant species, enabling insect species to co-evolve with particular plants. Research has found that the ratio of protein-to-carbohydrates (P:C) in a substance affects adult fly behavior, such as a low P:C ratio was associated with higher oviposition behaviors in D. suzukii, which is thought to be an evolutionary trait that enhances survivability in adult females [43]. These preferences, once fully characterized, can be used for targeted field surveillance, attract-and-sterilize, or attract-and-kill strategies.”  

Experiment sites (crop environments)

Point 3: Line 92 – why did the authors choose the environments that they did (cherimoya, lemon, strawberry)? Are these two flies pests in each of those crops and therefore are expected to be captured at each location?

Response 3: The original objective of this experiment only included SWD so a survey was conducted in several locations to identify populations of SWD. Survey sites were selected based on suspected sightings reported by farmers (cherimoya, lemon, banana) and known host-plants (strawberry). The results of the survey also identified Zaprionus indianus (African fig fly; AFF), which was previously unreported on the island and published in the Proceedings of the Hawaiian Entomological Society. Ultimately, the environments that were selected were chosen based on the results of that survey – the only requirement was that the environment must have populations of both AFF and SWD.

Point 4: I am not familiar with cherimoya but I imagine the rind of a lemon would be too thick and tough for either fly to penetrate without the lemon being damaged.

Response 4: The variety of lemon growing in Kula had a very thin and soft skin, unlike many of the thicker skin lemons that are sold in traditional supermarkets. As the SWD survey was being performed, I observed an exceptional population of drosophilids in the lemon grove – so many that the limiting factor for quantity of field captures was the volume of attractant/drowning solution (40 mL). Though SWD or AFF may not be a traditional pest of lemon fruit, the sheer quantity of drosophilids was concerning and I thought the results may be interesting. There was a survey performed in the late 1960’s to early 1970’s that reported using fresh sliced orange and grapefruit as baits and described citrus fruits as a major food source for drosophilids (Bryant et al., 1982; https://www.jstor.org/stable/2407963). I suspect that adult flies, particularly SWD and AFF, dispersed from the adjacent strawberry field as the fruit was harvested to the lemon grove where fruit was ripening. Cherimoya that is unripe and undamaged can be hard, but  as the fruit ripens it becomes soft, fragrant, juicy, and highly susceptible to bruising.     

Point 5: Strawberry is an important host plant of SWD and as the authors mention, can be for Zaprionus. However, the study was conducted post-harvest when fruit present was over-ripe. If SWD and Zaprionus prefer ripe strawberries (Bernardi et al. 2017) then would the authors expect for these traps to give an adequate picture of olfactory response post-season?

Response 5: It is quite disappointing that the experiment took place in a strawberry field post-harvest. All three experiments were designed to be run concurrently to control for wind, temperature, relative humidity and other factors that may have impacted the results. Since the strawberry fruit was ripe during the survey, it was selected as a survey site. Due to the fruit being so close to harvest, it was requested that the traps be placed on the perimeter of the field (as to not disrupt harvest) or to wait a week until post-harvest. In hindsight, I would have chosen to place the traps along the perimeter of a field of ripe fruit rather than within the field post-harvest. Had the former path been chosen, we could have also reviewed captures in the field pre-and post-harvest, which may have been quite interesting! I expected there to be enough SWD and AFF captures to analyze attractant trends in the field post-harvest (as there were numerous captures the week prior!). Also, the experiments in strawberry and lemon ended prematurely due to re-planting and harvesting, respectively.

Point 6: Replication over time ranged from 2 (lemon and strawberry) to 4 (cherimoya). Attractiveness of baits can vary over the season due to changes in nutritional needs and reproductive status of the flies. Based on the statistical models in this study, captures may have varied by attractant over time. Once the analysis is redone with the appropriate models the authors may want to evaluate whether 2 replications over time in lemon and strawberry are adequate and potentially omit those two environments.

Response 6: Prior to the re-analysis of data from ANOVA to GLMM, the lemon and strawberry host-plant environment results were going to be reported since it was emphasized in the discussion that the short duration was a limitation of the experimental design. After using the improved statistical model (GLMM), it became apparent that sparse data was an issue in the strawberry data set for both D. suzukii and Z. indianus. The high variability and low replication for lemon decreased confidence in interpreting the results of this data set. Therefore, strawberry and lemon were omitted from the analysis and further analysis of attractant specificity was focused on cherimoya.

Statistics

Point 7: I have some major concerns with the statistical model that was used. Based on the text, the authors used a two-way ANOVA with attractant, week(block), and attractant*week(block) as the fixed effects. First, unless the authors are interested in the effect of the attractant over time (the interaction) then the week(block) should be added as a random variable rather than a fixed variable. If the authors are interested in the effect of attractant over time then it would be considered a longitudinal study whereby each count was taken on the same trap and would need to be analyzed using repeated measures. Having only two time points in lemon and strawberry however, may not provide enough data for repeated measures analysis.

Response 7: The major concerns over the statistical model were warranted. Though the effect of time is interesting, the experiments were not a repeated measures design. Week was selected as a fixed effect as an attempt to block the effect of time. The data should have been run as a mixed model with week as a fixed effect. As suggested, the statistical model was re-designed as a GLMM with attractant type as a fixed effect with replicate and week as random effects. Additional details of the revised model can be found on response 9. 

Point 8: Next, why wasn’t replicate included as a random variable in the model? This would account for any variability that may be a result of the replication over space. This would also address what I believe the authors were testing for in lines 188-192, of positional bias of the traps within each environment.

Response 8: Replicate should have been included as a random variable in the ANOVA model and, for this revision, was added as a random effect in the GLMM.

Point 9: Finally, there have been a multitude of studies suggesting that count data is better modelled using a generalized linear model (PROC GLIMMIX for generalized linear mixed models in SAS). They are not new but are becoming the standard for modelling this type of data. The distribution function is defined in the model (eg. Poisson or Negative Binomial for count data) so no transformation is required. Additionally, the long final paragraph in the statistics section of the paper would not be necessary. See Bolker et al. 2009 https://doi.org/10.1016/j.tree.2008.10.008; Gbur et al. 2012 https://dl.sciencesocieties.org/publications/books/tocs/acsesspublicati/analysisofgener for starters

Response 9: The statistical model was completely changed in light of this review of the manuscript. PROC GLIMMIX was used with the distribution defined as Poisson (no transformation needed!). Random effects were defined as week and replicate. The GLMM was an excellent fit for the data in comparison to the previous ANOVA model. The results from the mean separation in post-hoc analysis appeared to correspond very well to the trends observed in graphical representations of the data – as it should! 

The final paragraph of the statistics section was removed from the revision.

Point 10: Additionally, there is no statistical comparison between male and female captures of SWD or Zaprionus or between SWD and Zaprionus. The authors cannot state that the results of their study suggest that these two pest species have unique olfactory preferences (lines 364-366) if they were not compared.

Response 10: In order to compare male and female differences between species and between host-plant environments, ANOVA was used to examine the effects of attractant within the host-plant environment and within species/sex category, the results of which were then compared to each other, which may be an indirect/unfair comparison. 

The results of the statistical comparison between male and female captures within the same species were added to the manuscript:

 “Since there were no significant differences between male and female Drosophila suzukii captures in cherimoya (f=0.73, df=4, 68, p=0.5761), captures were compiled for both sexes for analysis.”

“Since there were no significant differences between male and female Zaprionus indianus captures in cherimoya (f=1.43, df=4, 14, p=0.2762), captures were compiled for both sexes for analysis.”

Furthermore, specificity was calculated between D. suzukii and all drosophilids, between Z. indianus and all drosophilids, and between D. suzukii and Z. indianus. An example of how the data was manipulated and interpreted was added to the statistical analysis section:

 “In order to determine whether there were significant differences in mean captures between the sexes, the following formula was used to manipulate two columns of raw data into one response variable for analysis:

Sex specificity = a / (a + b)           (1)

where:

a = count of male captures

b = count of female captures

The sex specificity formula results in a value between 0 and 1 for each datapoint (row of the spreadsheet). A value of 1.0 implies that only males were captured, while a value of 0.5 implies that there was equal male to female captures, and a value of 0.0 indicates that only females were captured.

To examine whether there were sex differences in mean capture between attractants, a generalized linear mixed model (GLMM) was used (PROC GLIMMIX) with attractant as a fixed effect and with week and replicate as random effects. Method was set as Laplace and a beta distribution was specified as the sex specificity calculation was expressed as a decimal fraction. If there were no significant differences in sex specificity between either male and female D. suzukii or between male and female Z. indianus, further analysis was performed with the sex counts compiled. To support the primary objective, balanced analysis of variance (ANOVA) was performed (PROC ANOVA) with week as a blocking factor were conducted to assess the factor of attractant (five levels)

To examine differences in mean capture between attractants, a GLMM was used (PROC GLIMMIX) with attractant as a fixed effect and with week and replicate as random effects. Method was set as Laplace and a Poisson distribution was specified for the raw count measurements for each specimen category (D. suzukii female, D. suzukii male, Z. indianus female, Z. indianus male, non-target drosophilid, and other non-targets).

To examine specificity between D. suzukii v. drosophilid captures, Z. indianus v. drosophilid captures, and D. suzukii v. Z. indianus captures raw data was manipulated with the following formulas into one response variable for analysis:

D. suzukii v. drosophilid specificity = (a + b) / (a + b + c + d + e)         (2)

where:

a = count of D. suzukii male captures

b = count of D. suzukii female captures

c = count of Z. indianus male captures

d = count of Z. indianus female captures

e = count of non-target drosophilid captures

Z. indianus v. drosophilid specificity = (a + b) / (a + b + c + d + e)          (3)

where:

a = count of Z. indianus male captures

b = count of Z. indianus female captures

c = count of D. suzukii male captures

d = count of D. suzukii female captures

e = count of non-target drosophilid captures

D. suzukii v. Z. indianus specificity = (a + b) / (a + b + c + d) (4)

where:

a = count of D. suzukii male captures

b = count of D. suzukii female captures

c = count of Z. indianus male captures

d = count of Z. indianus female captures

 The specificity formulas results in a value between 0 and 1 for each datapoint (row of the spreadsheet) with interpretation of values similar to as previously described for formula (1).

To examine whether there were differences in specificity of captures between attractants, a generalized linear mixed model (GLMM) was used (PROC GLIMMIX) with attractant as a fixed effect and with week and replicate as random effects. Method was set as Laplace and a beta distribution was specified as the specificity results were expressed as decimal fractions.”

The results of the statistical comparisons of specificity were added to the manuscript in the following locations:

 “There were significant differences observed in the specificity rate of D. suzukii to all drosophilid captures in cherimoya (f=4.39, df=4,74, p>0.01) (Figure 4).”

“There were no significant differences observed in the specificity rate of Z. indianus to all drosophilid captures in cherimoya (f=1.82, df=4,37, p=0.1450).”

“There were no significant differences observed in the specificity rate of D. suzukii to Z. indianus captures in cherimoya (f=1.86, df=4,37, p=0.1387).”

Point 11: There was also no comparison among the different environments.

Response 11: Since strawberry and lemon were omitted due to experimental design limitations, this comment, though valid for the original manuscript, is no longer relevant for the current revision.

To reflect these changes in the experimental objectives, the end of the introduction was revised. This revision included removal of comparisons between host plant environments and added comparative specificity of attractants for the field capture of D. suzukii and Z. indianus as compared to each other and non-target drosophilids

Reviewer 2 Report

Drosophila Suzuki and Zaprionus indianus are serious invasive pests in US fruit orchards. This study performed field trap experiments to capture these two fruit flies by using odor attractants. The objectives of this study were to clarify efficacies of attractants in different host-plant environments and to characterize the relative abundance of the two flies in these environments on Maui, Hawaii. I feel the experiments were performed in a good manner and the results were clearly demonstrated.

However, I found potential incoherence in this manuscript. First, the authors described that “the effectiveness of a given compound to elicit an attraction response is known to be influenced by environmental cues and physiological adaptations (L427)” and concluded that “the data suggest that the host-plant environment affects attractant efficacy (L498)”, but the present study cannot rule out the possibility of the behavioral differences attributed to genetic differences of flies. The flies in different host-plant habitats may be in an initial step of a divergence of host races. Second, the primary objective and the secondary objective are inconsistent; if the files show different responses to traps in different environments in different species, the data from such the trap-based census cannot reflect the true relative abundance of each species. The data would underestimate the abundance of the species that show relatively weak responses.

I recommend the authors revising the manuscript very carefully.

Author Response

Reviewer 2 comments

Summary of manuscript: Drosophila Suzuki and Zaprionus indianus are serious invasive pests in US fruit orchards. This study performed field trap experiments to capture these two fruit flies by using odor attractants. The objectives of this study were to clarify efficacies of attractants in different host-plant environments and to characterize the relative abundance of the two flies in these environments on Maui, Hawaii. I feel the experiments were performed in a good manner and the results were clearly demonstrated.

Point 1: However, I found potential incoherence in this manuscript. First, the authors described that “the effectiveness of a given compound to elicit an attraction response is known to be influenced by environmental cues and physiological adaptations (L427)” and concluded that “the data suggest that the host-plant environment affects attractant efficacy (L498)”, but the present study cannot rule out the possibility of the behavioral differences attributed to genetic differences of flies. The flies in different host-plant habitats may be in an initial step of a divergence of host races.

Response 1: In response to the advisement of reviewer 1, the statistical model and the objectives of the manuscript have been modified to account for two of the host-plant environments (lemon and strawberry) being omitted from the report (due to low replication, high variability, and sparse data). Since the objectives no longer include comparisons between the three host-plant environments, the incoherence has been resolved in the revised manuscript. The discussion of species divergence is fascinating so a little more background was provided in the introduction below (in bold to emphasize additions):

“The concept of targeted attractants for drosophilid species is plausible as fermentation products are innately attractive to drosophilids [34-36] and drosophilids exhibit differences in olfactory preferences. Genetic and epigenetic traits regulate behavior and, over-time, as drosophilid populations compete over finite resources, these behaviors either result in no viable offspring, have no effect on viable offspring, or enhance the success and/or quantity of offspring. Generally, behaviors that are associated with no viable offspring will eventually become extinct in a population, while behaviors that have no effect or a positive effect on offspring will remain as traits within the population. Over-time, particularly with geographic isolation, these ecological shifts can dramatically shape the behavior, morphology, and genetics of populations. Often, research of these mechanisms is performed retroactively, after speciation, through a variety of methods, such as examination of olfactory preferences [37] behavioral experiments, and molecular analysis [38-40]. These species-specific preferences to different yeasts [41] may be related to evolutionary adaptations for survival against competing drosophilids over finite resources [23,42].

Point 2: Second, the primary objective and the secondary objective are inconsistent; if the files show different responses to traps in different environments in different species, the data from such the trap-based census cannot reflect the true relative abundance of each species. The data would underestimate the abundance of the species that show relatively weak responses.

Response 2: In order to shed light on this interesting point, the discussion has been revised.

The following text has been added (added text in bold font to emphasize changes):

The results of this experiment beg the question – if specificity varies depending on attractant used, how can an accurate representation of relative abundance be depicted? In this experiment, the relative abundance of D. suzukii and Z. indianus ranged from 22.3-46.4% and 1.1-5.8%, respectively, depending on the attractant used. Since longitudinal characterization of relative abundance is necessary to document the invasion status of an introduced species, and any corresponding effects to endemic species and the local economy over time, the accuracy of these calculations is vital. The results of this experiment emphasize the importance of documenting attractant specificity and using a wide variety of attractants to increase the accuracy of estimated relative abundance of insect populations. The relative abundance in this experiment was calculated by totaling captures across all attractants, leading to an estimated relative abundance of captured drosophilids as 34.3% D. suzukii, 2.9% Z. indianus, and 62.8% for non-target drosophilids. The high relative abundance of D. suzukii suggests that cherimoya may be a preferred host or that the lack of overripe fruit on the ground (well-managed grove) discouraged population expansion of other drosophilids. The low relative abundance of Z. indianus suggests that either cherimoya is not a preferred host, or that the attractants used were not more attractive than the host-plant environment. – which may underestimate the true relative abundance. Adult Z. indianus specimens are often captured in attractants used to monitor D. suzukii (i.e., wines and vinegars), but there may be more effective attractants for Z. indianus. Grape juice has been demonstrated to be an effective attractant for Z. indianus [58], which was not assessed in these experiments. The use of grape juice, or other attractants that may have higher specificity for Z. indianus, would have resulted in higher, perhaps more accurate, estimates of relative abundance.  However, the low relative abundance of Z. indianus may also be attributed to the recent introduction of this species to the region [47]. In central Brazil, Tidon et al. [59] reported that Z. indianus had an initially low relative abundance followed by a population explosion the following year when populations were re-surveyed. Furthermore, Tidon et al. [59] observed that Z. indianus specimens were more prevalent in savanna sites, suggesting that the microclimates in lower Kula and central Maui may support greater densities of Z. indianus than the low relative abundance that was observed in upper Kula for this experiment.

Summary of comments: I recommend the authors revising the manuscript very carefully.

Reviewer 3 Report

The authors should review all the results section:

Please, check the standard error for all the values provided on the text or in the Figures.

In my opinion it would be more informative (and probably showing significant differences) to show the dynamic population for each species or group of insects in each kind of crop and analyze the effect of the traps for each week separately.

Please provide the statics to justify the analyses of males and females separately if necessary.

Please, include the control in case of some capture.

Author Response

Reviewer 3 comments

The authors should review all the results section:

Point 1: Please, check the standard error for all the values provided on the text or in the Figures.

Response 1: For this revision, all of the figures were re-generated based on analysis of the data as a GLMM instead of an ANOVA. After the revisions were finalized, the averages and standard errors were carefully inspected in both the text and the figures.

Point 2: In my opinion it would be more informative (and probably showing significant differences) to show the dynamic population for each species or group of insects in each kind of crop and analyze the effect of the traps for each week separately.

Response 2: This would be quite interesting. Therefore, the following changes were made:

The objective in the introduction was revised to include the effect of time:

“The secondary objective was to examine whether attractant efficacy varied over time in the field capture of D. suzukii, Z. indianus, and non-target drosophilids.”

The following text was added to the methods:

“To support the secondary objective, a GLMM was used (PROC GLIMMIX) to examine whether mean captures or specificity varied by week (fixed effects= attractant, week, and attractant*week; random effect= replicate). Similar to the previous analysis, the method was set as Laplace and the distribution was set as Poisson and beta for count data and decimal fractions, respectively.”

The following results were added:

“There were significant differences observed in mean captures of D. suzukii by week surveyed (f=25.93, df=3,60, p<0.0001). The greatest mean captures of non-target drosophilids occurred in weeks 2 (24.5 ± 2.8) and 3 (22.0 ± 3.4), followed by week 1 (14.3 ± 2.7), with the lowest captures observed in week 4 (7.5 ± 1.0).”

“There were no significant differences observed in mean captures of Z. indianus by week (f=2.74, df=3,60, p=0.0511). or by the interaction of attractant and week (f=1.81, df=12,60, p=0.0671).”

“There were significant differences observed in mean captures by week surveyed (f=37.25, df=3,60, p<0.0001) (Figure 6). The greatest mean captures of non-target drosophilids occurred in week 3 (52.8 ± 6.8), followed by week 2 (36.4 ± 4.4), week 1 (23.0 ± 4.6), and the lowest were observed in week 4 (12.7 ± 1.7). Since no significant differences were observed between the interaction of week and attractant (f=1.44, df=12, 60, p=0.1748), no further analysis was performed. “

If the interaction between attractant and week was significant, the effect of attractant was analyzed for each week separately.

The following was added to the results:

“Since there were significant differences observed in mean captures by the interaction of week and attractant, (f=3.43, df=12,60, p<0.001), attractant preferences were examined separately for each week. Significant differences were observed in mean captures by attractant in week 1 (f=6.13, df=4,15, p<0.01), week 2 (f=6.73, df=4,15, p<0.01), week 3 (f=4.64, df=4,15, p<0.05), and week 4 (f=5.76, df=4,15, p<0.01) (Figure 5). For week 1, the use of BRV+RW (33.5 ± 4.25) resulted in more captures than BRV (11.5 ± 5.7), RW (10.0 ± 3.0), ACV+RW (9.5 ± 2.9), and ACV (6.75 ± 1.8). For week 2, the use of BRV+RW (32.0 ± 4.5), ACV+RW (31.8 ± 7.3), BRV (30.0 ± 5.6) or RW (19.3 ± 2.5), resulted in more captures than ACV (9.5 ± 3.5). Week 3 was similar to week 1 – the use of BRV+RW (43.8 ± 10.1) resulted in more captures than ACV (19.3 ± 5.7), ACV+RW (16.8 ± 4.5), BRV (16.5 ± 3.2), and RW (13.5 ± 2.5). In contrast, for week 4, ACV+RW (12.5 ± 1.3) resulted in more captures than BRV+RW (6.75 ± 1.6) and ACV (1.75 ± 0.5). The use of RW (9.25 ± 1.9), BRV (8.0 ± 2.7), or BRV+RW (6.75 ± 1.6) resulted in more captures than ACV (1.75 ± 0.5). There were no significant differences between ACV+RW (12.5 ± 1.3), RW (9.25 ± 1.9), and BRV (8.0 ± 2.7). There were no significant differences between RW (9.25 ± 1.9), BRV (8.0 ± 2.7), and BRV+RW (6.75 ± 1.6).”

Point 3: Please provide the statics to justify the analyses of males and females separately if necessary.

Response 3: In the re-analysis of the data as a GLMM, male and female captures were examined to see whether there were significant differences, if not, then the counts were compiled for both sexes.

The following text was added to the methodology section to describe the statistical testing between male and female captures:

“To examine whether there were sex differences in mean capture between attractants, a generalized linear mixed model (GLMM) was used (PROC GLIMMIX) with attractant as a fixed effect and with week and replicate as random effects. Method was set as Laplace and a beta distribution was specified as the sex specificity calculation was expressed as a decimal fraction. If there were no significant differences in sex specificity between either male and female D. suzukii or between male and female Z. indianus, further analysis was performed with the sex counts compiled.”

The results of the statistical comparison between male and female captures within the same species were added to the manuscript:

 “Since there were no significant differences between male and female Drosophila suzukii captures in cherimoya (f=0.73, df=4, 68, p=0.5761), captures were compiled for both sexes for analysis.”

“Since there were no significant differences between male and female Zaprionus indianus captures in cherimoya (f=1.43, df=4, 14, p=0.2762), captures were compiled for both sexes for analysis.”

Point 4: Please, include the control in case of some capture.

Response 4: The control captures were listed in the contingency table before the discussion, but the reported captures were not discussed at all in the text. Therefore, the control captures were reported and interpreted in two location in the discussion.

The following text was added to the first paragraph in the discussion:

“Since there were relatively few captures in the control treatment (distilled water), which controlled for drosophilid attraction towards visual stimuli (to the red solo cup) [31], it can be concluded that captures in this experiment were due to olfactory attraction or the combination of olfactory and visual stimuli – not to visual stimuli alone.”

The following text was added to the final paragraph in the discussion:

“Over the four-week experiment, there were few total captures for traps treated with distilled water, three D. suzukii specimens were captured (one male, two female), zero Z. indianus specimens were captured, five non-target drosophilids were captured, and nine non-target other specimens were captured. Due to the low quantity of control captures, there is confidence that the dataset analyzed contained values that were the result of olfactory attraction and not confounded by visual stimuli. However, the experiment did not examine the interaction or synergy between visual and olfactory stimuli. Additional research would be necessary to assess whether an attractant is more or less attractive in different visual environments.”

Round 2

Reviewer 1 Report

Comments for resubmitted manuscript

Overall, the authors have focused more on a single cropt (chemoya) and revised the statistical model. They have addressed all of my previous comments and I feel the paper is much clearer. I have just a couple final comments:

L83 – plant should be either plants or the plant, replace the second “behavior” in this sentence with “subsequent” or another rword

L122 – add fly in front of entry

L144 – replace “-“ with a comma

Statistics: make sure to define the level at which you are considering your analyses significant

Equations: you may be able to explain these equations in words such as the total number of SWD males and females divided by the total number of drosophilids captured in the trap”. If you want to keep them I would suggest using the same variable definitions for all equations. So for eq. 2 SWD male and female should still be a and b, respectively and the equation would look like (c+d)/(a+b+c+d+e)

L240-243 – since these are the same as eq. 1 and 2 you could mention that for clarification

L428 – add “in” before a laboratory trapping assay

L472 – limit should be limited

Author Response

Please review the attached word document, as each response contains screenshots from the revised manuscript. 

Response to reviewer 1 comments (round 2)

Overall, the authors have focused more on a single crop (cherimoya) and revised the statistical model. They have addressed all of my previous comments and I feel the paper is much clearer. I have just a couple final comments:

Point 1: L83 – plant should be either plants or the plant, replace the second “behavior” in this sentence with “subsequent” or another word

Response 1: Plant has been changed to plants and additional was replaced with subsequent, resolving the grammatical error and improving flow. The changes to the text are illustrated below.

Point 2: L122 – add fly in front of entry

Response 2: The word fly has been added in front of entry as illustrated below.  

Point 3: L144 – replace “-“ with a comma

Response 3: The use of “–“ has been replaced with a comma.

Point 4: Statistics: make sure to define the level at which you are considering your analyses significant

Response 4:  To correct this error of omission, a statement was added to the statistical analysis section, defining the statistical threshold.

Point 5: Equations: you may be able to explain these equations in words such as the total number of SWD males and females divided by the total number of drosophilids captured in the trap”. If you want to keep them I would suggest using the same variable definitions for all equations. So for eq. 2 SWD male and female should still be a and b, respectively and the equation would look like (c+d)/(a+b+c+d+e)

Response 5: The equations were added as an attempt to reduce ambiguity for the readers, but the calculations are simple enough to be described with words. Therefore, the formulas were omitted and replaced with text describing the methods used. The two images below illustrate the changes that were made to the text

Point 6: L240-243 – since these are the same as eq. 1 and 2 you could mention that for clarification

Response 6: For clarification, a statement was added to the text to describe the relationship between the estimated relative abundance and specificity calculations.

Point 7: L428 – add “in” before a laboratory trapping assay

Response 7: The word “in” was added to correct the sentence.  

Point 8: L472 – limit should be limited

Response 8: The word “limit” was replaced with “limited” to correct the sentence.

Reviewer 2 Report

I feel the manuscript is improved now. However, I would like to indicate one concern in the statistical analyses. The authors analyzed their whole data by GLMM but subsequently performed pair-wise comparisons by LSD. This is very strange. LSD is not a valid method for most biological and ecological data and is never recommended in our field. I think the authors should perform pair-wise comparisons by using GLMM followed by adjustment of p values with e.g. the Bonferroni method to control the Type I error in multiple comparisons.

Author Response

Please review the attached word document, as each response contains screenshots from the revised manuscript. 

Response to reviewer 2 comments (round 2)

Point 1: I feel the manuscript is improved now. However, I would like to indicate one concern in the statistical analyses. The authors analyzed their whole data by GLMM but subsequently performed pair-wise comparisons by LSD. This is very strange. LSD is not a valid method for most biological and ecological data and is never recommended in our field. I think the authors should perform pair-wise comparisons by using GLMM followed by adjustment of p values with e.g. the Bonferroni method to control the Type I error in multiple comparisons.

Response 1: My apologies, the way in which the text and figure captions were written has led to a misunderstanding. I did not describe the post-hoc analysis in great enough detail and should have used the phrase “LS means” in the statistical section, rather than listing it in figure captions following the p-value threshold where Tukey’s HSD should have been listed.

LSD was not used as a post-hoc test, rather Tukey’s test was used. When performing post-hoc analysis in SAS for a GLMM (PROC GLIMMIX), two statements can be used: MEANS or LSMEANS. MEANS calculates the arithmetic mean, while LSMEANS is used in cases where there is an unbalanced design (this was not) or in mixed models to account for covariates (this model controlled for replicate and week as random effects). Mean separation was then performed using Tukey’s HSD test.  

To clarify the text, the statistical analysis section and figure captures were updated as follows:

Reviewer 3 Report

The manuscript improved consderably from the first version. However I think that you should work more in the structure (mostly in the result sections) and rewrite the discussion sections trying to do not repeat the results if there is nothing to discuss about.

I give some suggestions to keep improving the manuscript.

Figure 3, 4 and 5. I think that with figure 4 and 5 you are already showing the results that you want to show. I recommend discussing that in the peak of D. suzukii population, the different traps do not have significant differences.

I suggest reducing the number of sections in the results. I would put both invasive pest species in the same section. First, you could compare the populations dynamics (even if Z. indianus do not have significant differences). And then show the figure for the captures by week and attractant for both species.

I suggest also to merge the sections 3.2.1. and 3.2.2. in one.

Section 3.2.3. Eliminate this section. You can not compare the specificity of both species if you do not know the initial population in the field.

Line 19: Eliminate the “,” after ACV. I suggest to re-write that sentence in this way: BRV+RW and ACV+RW resulted in more captures than BRV, ACV and RW.

Line 183: After eliminating the others crops not make sense to me to keep the map of the experimental location (Figure 2).

Line 191 to 199: That is the definition of the sex ratio. Please consider do not write the formula.

Line 214 to 219: That is the ratio of D. suzukii respect the rest of the drosophilids.

Line 220 to 225: That is the ratio of Z. indianus respect the rest of the drosophilids. Please consider do not write the formula.

Line 226 to 230: That is the ratio of D. suzukii respect the total of the invasive species individuals

Line 240 to 241: Is it the same number than the defined previously (Lines 214 to 225)?

Line 247 to 248: I am not sure that is necessary to define sex ratio in material and methods if after all there are no differences between males and females for both species. Please, consider to discard the explanation of the sex ratio in material and methods or explain briefly and write at the beginning of the results section.

Line 250: Why not to use the standard error?

Lines 251 to 256: Please consider to rewrite these lines organizing by the three groups of efficacy that we can observe.

Line 261: It seems interesting than there were more captures in week two and three. Maybe it will be interesting to correlate these values with the mean temperature of the area each week.

Figure 5: Please keep the order of the attractant types for all the figures.

Lines 287 to 293: Please, expand this result section since there is no figure to view the data, for example how many insects you catch each week? The total amount per attractant type?

Line 296: change “or” to “and”

Line 297: change “or” to “and”

Line 298 and 299: repetitive

Line 304: change figure 6 to figure 7 in the text

Line 316: change figure 4 to figure 8

Line 314 to 320: It seems that all the BRV traps (with and without RW) are more specific for D. suzukii (almost 50 % of the insects where D. suzukii). I suggest remarking this results.

I suggest to eliminate the result section 3.2.2. and 3.2.3 (Maybe to maintain the structure of the paper and write down this results you could merge the results for each species or groups)

I consider that Figure 9 it is not necessary since there are not the previous comparisons between crops. You can write about this data just in the text.

Table 1. You said that there were not significant differences between sexes for D. suzukii and Z. indianus. However, I can see a clear pattern on Table 1. For all the attractant solutions you are catching almost the double of males than females for D. suzukii. Please consider to reanalyze the data by total catches per attractant solution and discuss about the number of males and females in the field. Do that result reflects the sex ratio of the population in the field?

Line 341 and 342: rewrite statement (2) D. suzukii has a preference … . The data is at local scale in only one crop.

Line 340 to 345: Are these statements the answers to the questions addressed in the objectives of the introduction?

Line 356: It would be good to put a reference about the importance of these studies. Maybe a review?

Line 363: A reference is needed.

Line 365 to 369: It sound repetitive to me. Please, merge both references in one sentence.

Line 380: So I would conclude that is better to use the more specific attractant for D. suzukii 

Line 385: for Z. indianus or D. suzukki. Both?

Line 388 to 393: This is the discussion section. Please do not repeat the results or move this paragraph to results section.

Line 402 to 404: repetitive. A reference is needed or eliminate  “which is thought to be the result of evolved olfactory preferences for oviposition and feeding”.

Line 422 to 423: The results of this manuscript do not support this statement. Please eliminate.

Line 432 to 433: Please do not repeat results on the discussion section if there is not discussion.

Line 446 to 449: This could be solved with a very simple experiment in the lab or in the greenhouse.

Line 457 to 460: I don’t understand what is the relation of a Z. indianus population in central Brazil and the ones in two different locations from Hawai. Please develop more this discussion.

Line 468: delete the point

Line 470: where? In Virginia or in Hawai? Both?

Line 499 to 500: How you know if there was or not a high specificity of the attractants or low population level?. I think that this conclusion can not be claimed.

Author Response

Please review the attached word document, as screenshots are used to illustrate revisions. 

Response to reviewer 3 comments (round 2)

The manuscript improved consderably from the first version. However I think that you should work more in the structure (mostly in the result sections) and rewrite the discussion sections trying to do not repeat the results if there is nothing to discuss about.

I give some suggestions to keep improving the manuscript.

Point 1: Figure 3, 4 and 5. I think that with figure 4 and 5 you are already showing the results that you want to show. I recommend discussing that in the peak of D. suzukii population, the different traps do not have significant differences.

Response 1: Although figures 3, 4, and 5 are somewhat similar, they reflect the results of the statistical comparisons that were performed. The objective of the experiment was to examine differences in mean captures (Figure 3), then it was revised to also include week (Figure 4), and since the result was significant, it required further analysis (Figure 5). To remove any of the figures would detract from either the objective or the story.  It is true that the figures could be replaced with words, but the information may be more easily comprehended with figures.

In the peak of the D. suzukii population, weeks 2 and 3, significant differences between attractants are observed. In week 2, BRV, BRV+RW, ACV+RW, and RW were more attractive than ACV, while in week 3, BRV+RW was more attractive than the other solutions. This was originally mentioned in the discussion, but was removed in point 34.

Point 2: I suggest reducing the number of sections in the results. I would put both invasive pest species in the same section. First, you could compare the populations dynamics (even if Z. indianus do not have significant differences). And then show the figure for the captures by week and attractant for both species.

Response 2: The number of subsections in the results was reduced. The results section was restructured to report all results by species.

Point 3: I suggest also to merge the sections 3.2.1. and 3.2.2. in one.

Response 3: The subheadings 3.2.1 and 3.2.2 were removed. The results section was restructured to report all results by species.

Point 4: Section 3.2.3. Eliminate this section. You can not compare the specificity of both species if you do not know the initial population in the field.

Response 4: While I disagree with this interpretation since the calculation for specificity did not include an estimation of the initial population, only the observed capture rate, it is important to error on the side of caution so the specificity calculation between D. suzukii and Z. indianus was removed. 

Furthermore, it is important to emphasize to the readers that the experiment occurred in an open-field, that the true population was unknown, and additional research is needed in a controlled environment. The following text was added to clarify.

Point 5: Line 19: Eliminate the “,” after ACV. I suggest to re-write that sentence in this way: BRV+RW and ACV+RW resulted in more captures than BRV, ACV and RW.

Response 5: The oxford comma was intentionally added and used consistently throughout the text in an attempt to be extra cautious regarding interpretation of attractant factor levels. Plus sides were added to the labels for mixed solutions, such as “ACV+RW” instead of “ACV and RW,” to reduce ambiguity as to whether the label is referring to mixed solutions. Even though these labels were added, omission of the oxford comma may lead to misinterpretation as the use of “ACV and RW” at the end of a sentence could be interpreted by readers as a mixed solution, particularly if the text is skimmed.

Furthermore, mean captures with ACV+RW were not significantly different from BRV or RW, only ACV.

Since there were only 193/200 words used in the abstract and text could be added to clarify the relationship observed, the sentence was split and an additional phrase added:

Point 6: Line 183: After eliminating the others crops not make sense to me to keep the map of the experimental location (Figure 2).

Response 6: The district of Kula has a very wide range of elevations and microclimates. Locals often use a prefix of “upper” and “lower” in an attempt to distinguish the elevation and climactic differences, even still, the elevation within the upper and lower boundaries varies quite a bit. Community labels within the district of Kula are frequently used by locals as well, such as Keokea, Omaopio, Pulehu, Poli Poli, and so on. As a resident, the term Kula, even when paired with an elevation, does not hold much meaning, while viewing the map can tell me a lot about the area. The use of Figure 2 provides a much better illustration of the area than I could describe to the readers with words.   

Point 7: Line 191 to 199: That is the definition of the sex ratio. Please consider do not write the formula.

Response 7: The formulas were added to reduce ambiguity, but the calculations are simple and can be described with words. To address this, the formula was omitted and the sex ratio was described in the text as illustrated below:  

Point 8: Line 214 to 219: That is the ratio of D. suzukii respect the rest of the drosophilids.

Response 8: The formula was removed and replaced with text describing the calculation.

The following text was added: 

The following text was removed:

Point 9: Line 220 to 225: That is the ratio of Z. indianus respect the rest of the drosophilids. Please consider do not write the formula.

Response 9: The formula was removed and replaced with text describing the calculation.

The following text was added: 

The following text was removed:

Point 10: Line 226 to 230: That is the ratio of D. suzukii respect the total of the invasive species individuals

Response 10: The formula was removed and replaced with text describing the calculation.

The following text was added: 

The following text was removed:

Point 11: Line 240 to 241: Is it the same number than the defined previously (Lines 214 to 225)?

Response 11: No, though they are similar calculations. The estimation of relative abundance takes into consideration all of the attractant levels, while the purpose of the specificity calculation is to provide a single response variable for each replicate to permit statistical analysis. An additional statement was added to clarify.

Point 12: Line 247 to 248: I am not sure that is necessary to define sex ratio in material and methods if after all there are no differences between males and females for both species. Please, consider to discard the explanation of the sex ratio in material and methods or explain briefly and write at the beginning of the results section.

Response 12: Since the calculations and statistical analyses were performed in the experiment, excluding this methodology, simply due to the lack of significance in the results, seems unusual.  The calculation of these values was changed from a formula to a text as described in the responses to points 7 through 10, which resulted in a briefer explanation. 

Point 13: Line 250: Why not to use the standard error?

Response 13: In the results section, the text reporting D. suzukii captures by attractant was updated with the SEM values, instead of the SD, to correspond with the figures. The changes are illustrated below.

Point 14: Lines 251 to 256: Please consider to rewrite these lines organizing by the three groups of efficacy that we can observe.

Response 14: The lines were re-written or removed in order to reduce redundancy and increase clarity, as illustrated below.   

Point 15: Line 261: It seems interesting than there were more captures in week two and three. Maybe it will be interesting to correlate these values with the mean temperature of the area each week.

Response 15: Correlating the capture values with the mean temperature each week would be quite interesting. Unfortunately, I did not record these values during the experiment. A small weather station should have been used in the grove to record temperature, relative humidity, precipitation, and wind speed. I attempted to retrieve temperature data from other weather stations, but I am not confident that that it accurately represents the area in which the experiment occurred. Since Kula represents a wide range of elevations (differences of ~2,000 feet) and microclimates, it ranges quite a bit. I calculated the mean temperature reported from US Climate Data (https://www.usclimatedata.com/climate/kula/hawaii/united-states/ushi0052/2017/11) for Kula which resulted in mean weekly temperatures of 69, 68, 68, and 68°F for weeks 1, 2, 3, and 4, respectively. There is definitely more to the story, but it may not be possible to retrieve accurate data at this point. Methodology for future experiments will certainly include these measurements.  

Point 16: Figure 5: Please keep the order of the attractant types for all the figures.

Response 16: Figure 5 was corrected by re-ordering the attractant values as [ACV, BRV, RW, ACV+RW, and BRV+RW] to correspond with the ordering in all other figures.

Point 17: Lines 287 to 293: Please, expand this result section since there is no figure to view the data, for example how many insects you catch each week? The total amount per attractant type?

Response 17: The contingency table lists the total amount of Z. indianus captures per attractant type, but does not report total captures by week. The results section was expanded to include figures for non-significant results for Z. indianus captures by attractant type (Figure 7), by week (Figure 8), and specificity (Figure 9). 

Point 18: Line 296: change “or” to “and”

Response 18: Changed “or” to “and”

Point 19: Line 297: change “or” to “and”

Response 19: Changed “or” to “and”

Point 20: Line 298 and 299: repetitive

Response 20: Lines deleted.

Point 21: Line 304: change figure 6 to figure 7 in the text

Response 21: Figures were re-numbered according to placement within the revised results section

Point 22: Line 316: change figure 4 to figure 8

Response 22: Figures were re-numbered according to placement within the revised results section

Point 23: Line 314 to 320: It seems that all the BRV traps (with and without RW) are more specific for D. suzukii (almost 50 % of the insects where D. suzukii). I suggest remarking this results.

Response 23: A statement was added to the discussion to draw more attention to this result.

Point 24: I suggest to eliminate the result section 3.2.2. and 3.2.3 (Maybe to maintain the structure of the paper and write down this results you could merge the results for each species or groups)

Response 24: The subheadings 3.2.2 and 3.2.3 were removed. The results section was restructured to report all results by species.

Point 25: I consider that Figure 9 it is not necessary since there are not the previous comparisons between crops. You can write about this data just in the text.

Response 25: It is not necessary, but as a reader I would rather see a pie chart depicting the results of one of the objectives than to have to skim the paper to find percentages.

Point 26: Table 1. You said that there were not significant differences between sexes for D. suzukii and Z. indianus. However, I can see a clear pattern on Table 1. For all the attractant solutions you are catching almost the double of males than females for D. suzukii. Please consider to reanalyze the data by total catches per attractant solution and discuss about the number of males and females in the field. Do that result reflects the sex ratio of the population in the field?

Response 26: Even though there were more male than female captures overall (~65 v. ~35%), the proportion of captures by attractant type were not significantly different (60-67% v. 33-40%). Since attractant preferences did not vary by sex, captures were compiled.

To clarify that what was examined was differences in attractant preferences by sex and not whether there were differences in total raw captures, the following text was revised.

A few sentences were added to the discussion to emphasize the higher population of males to females.  

Point 27: Line 341 and 342: rewrite statement (2) D. suzukii has a preference … . The data is at local scale in only one crop.

Response 27: The beginning of the statement, which precedes the findings, specifies that the results are limited to one crop, but it did not specificity locality. The statement has been modified to include locality, as illustrated below.

To further emphasize that the manuscript discusses one island in one host-plant, the following statement was added:

Point 28: Line 340 to 345: Are these statements the answers to the questions addressed in the objectives of the introduction?

Response 28: Yes, the first sentence of the discussion was meant to briefly summarize the findings of the objectives and to outline the remainder of the text. Since the revised objective of the interaction of attractant*time was not included in this statement, a brief statement was included, as indicated below.

Point 29: Line 356: It would be good to put a reference about the importance of these studies. Maybe a review?

Response 29: A couple of interesting studies were added as references to the line suggested.

Point 30: Line 363: A reference is needed.

Response 30: References added to the line, which are discussed in a greater detail in the lines that follow.

Point 31: Line 365 to 369: It sound repetitive to me. Please, merge both references in one sentence.

Response 31: The first sentence summaries both experiments on the level of “wine” and “vinegar” stressing that blends are typically more attractive than individual components, while the following two lines describe each of the findings in greater detail on the level of what type of wine and what type of vinegar was examined. It may sound repetitive because the words vinegar and wine are used frequently. We can merge the two sentences into one sentence and abbreviate RW and ACV, as illustrated below.

Point 32: Line 380: So I would conclude that is better to use the more specific attractant for D. suzukii

Response 32: Additional statements were added to the paragraph regarding specificity to address this point.

Point 33: Line 385: for Z. indianus or D. suzukki. Both?

Response 33: The sentence that runs from line 382-385 refers to Z. indianus

The sentence that runs from line 385-387 refers to D. suzukii

Point 34: Line 388 to 393: This is the discussion section. Please do not repeat the results or move this paragraph to results section.

Response 34: Removed the lines discussing the specific differences in captures by week, as it is already listed in the results.

Point 35: Line 402 to 404: repetitive. A reference is needed or eliminate  “which is thought to be the result of evolved olfactory preferences for oviposition and feeding”.

Response 35: Removed sentence. This was already discussed in the introduction.  

Point 36: Line 422 to 423: The results of this manuscript do not support this statement. Please eliminate.

Response 36: Removed sentence.

Point 37: Line 432 to 433: Please do not repeat results on the discussion section if there is not discussion.

Response 37: Removed sentence.

Point 38: Line 446 to 449: This could be solved with a very simple experiment in the lab or in the greenhouse.

Response 38: This would yield interesting results. A sentence was added to the following line to list a potential objective statement for such an experiment.

Point 39: Line 457 to 460: I don’t understand what is the relation of a Z. indianus population in central Brazil and the ones in two different locations from Hawai. Please develop more this discussion.

Response 39: Additional text was added to the discussion to clarify.  

Point 40: Line 468: delete the point

Response 40: Instead of deleting the fragmented sentence, it was corrected. Additional sentences were added to enhance the paragraph on co-infestation. 

Point 41: Line 470: where? In Virginia or in Hawai? Both?

Response 41: Removed the term “local” to emphasize that all farmers need practical and evidence-based strategies.

Point 42: Line 499 to 500: How you know if there was or not a high specificity of the attractants or low population level?. I think that this conclusion can not be claimed.

Response 42: The conclusion states what was observed (estimated low relative abundance) and then stresses the possibility that the value could have been underestimated due to the attractants used, so it was implied that the original conclusion could also be true (actual low relative abundance). To clarify the argument, additional statements were added stating that the low relative abundance observed could simply indicate a low population level.  

Insects EISSN 2075-4450 Published by MDPI AG, Basel, Switzerland RSS E-Mail Table of Contents Alert
Back to Top